# Power Lines: Scaling Laws for Weight Decay and Batch Size in LLM Pre-training

**Shane Bergsma, Nolan Dey, Gurpreet Gosal, Gavia Gray, Daria Soboleva, Joel Hestness**
Cerebras Systems
{shane.bergsma,joel}@cerebras.net

## Abstract

Efficient LLM pre-training requires well-tuned hyperparameters (HPs), including learning rate $\eta$ and weight decay $\lambda$. We study *scaling laws* for HPs: formulas for how to scale HPs as we scale model size $N$, dataset size $D$, and batch size $B$. Recent work [1] suggests the AdamW timescale, $\tau = B/(\eta\lambda D)$, should remain constant across training settings, and we verify the implication that optimal $\lambda$ scales linearly with $B$, for a *fixed $N$ and $D$*. However, as $N$ and $D$ *scale*, we show optimal $\tau$ obeys a precise power law in the tokens-per-parameter ratio, $D/N$. This law thus provides a method to accurately predict $\lambda_{\text{opt}}$ in advance of large-scale training. We also study scaling laws for optimal batch size $B_{\text{opt}}$ (the $B$ enabling lowest loss at a given $N, D$) and critical batch size $B_{\text{crit}}$ (the $B$ beyond which further data parallelism becomes ineffective). In contrast to prior work, we find both $B_{\text{opt}}$ and $B_{\text{crit}}$ scale as power laws in $D$, independent of model size, $N$. Finally, we analyze how these findings inform the real-world selection of Pareto-optimal $N$ and $D$ under dual training time and compute objectives.

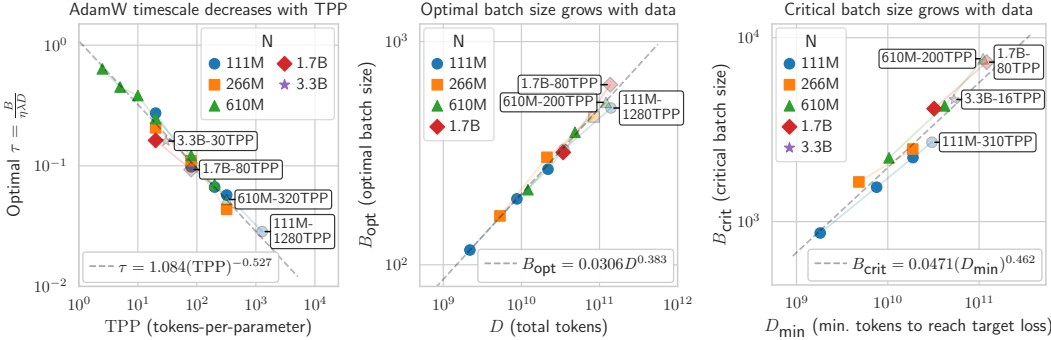

Figure 1: **Hyperparameters and their power lines**: Optimal $\tau$ obeys a power law in tokens-per-parameter (*left*), while optimal batch size (*middle*) and critical batch size (*right*) obey power laws in $D$. Faded markers indicate points not used in fitting; all fits generalize well to larger-scale runs.

## 1 Introduction

LLMs predictably improve as model size $N$ and training data size $D$ increase [2–4]. Today, state-of-the-art LLMs are trained at computational scales that leave no scope for hyperparameter (HP) tuning, although it is widely accepted that good HPs are critical for effective training [5–7].

Both theoretical and empirical efforts have sought to address this. Theoretically, maximal update parameterization ($\mu$P) allows the optimal learning rate $\eta_{\text{opt}}$ and initial weight variance $\sigma_{\text{opt}}^2$ to remain

39th Conference on Neural Information Processing Systems (NeurIPS 2025).

stable when scaling model width [8, 5], enabling a "tune small and train large" strategy. Empirically, DeepSeek LLM [7] adopted "scaling laws for HPs," where optimal batch size $B_{\text{opt}}$ and optimal learning rate $\eta_{\text{opt}}$ are estimated at small scale, and then extrapolated via a power law fit in total compute FLOPs, $C$. A similar approach was used in Kaplan et al. [4], forecasting $B_{\text{opt}}$ and $\eta_{\text{opt}}$ from loss $L$ and model size $N$.

Relying on a unique predicted $B_{\text{opt}}$ is *inflexible*—it precludes adjusting $B$ for compute/time trade-offs or hardware constraints. It is also unclear whether $C$, $L$, $N$, or $D$ (or a combination) best explains scaling. [7] notes, "for models with the same [$C$] but different model/data allocations, the optimal parameter space varies slightly." Also, no comparable study has been done for weight decay $\lambda$.

This paper introduces a flexible, unified approach to HPs, using both $\mu$P and scaling laws. We fit power laws to losses derived from hundreds of $\mu$P-trained models, focusing on combinations of $\lambda$, $B$, $N$, and $D$. We study both *compute-optimal* and *overtrained* models. The fewest FLOPs to achieve a loss typically occurs when training at $\approx$20 tokens-per-parameter (TPP = $D/N$) [3, 9], but overtrained models ($>$20 TPP) offer more-efficient inference [10]. We study TPPs from 20 to 1280.

To capture $\lambda$'s interaction with other scaling HPs ($\eta$, $B$), we model scaling of the AdamW timescale, $\tau = B/(\eta\lambda D)$. [1] found optimal $\tau$ stable with varying $D$, but we show it obeys a power law in TPP (Fig. 1, *left*). This law thus enables accurate estimation of $\lambda_{\text{opt}}$ for any $N$, $D$, $B$.

Leveraging these better HPs as $B$ scales, we also study optimal batch size: the $B_{\text{opt}}$ that minimizes loss at a given $N$ and $D$. While $B_{\text{opt}}$ scales as a power law in $C$ *when TPP is fixed* (Fig. 5, *left*), our results show this arises from a more fundamental power-law dependence on $D$ (Fig. 1, *middle*).

Importantly, increasing $B > B_{\text{opt}}$ can still reduce training time (fewer steps) and improve hardware utilization. This raises the question: how much *extra* data is needed when using large $B$? Prior work defines the *critical batch size* $B_{\text{crit}}$ as the point where training to a target loss requires $2 \times D_{\text{min}}$, with rapidly-diminishing returns in training speed thereafter [11]. We show $B_{\text{crit}}$ also scales with $D$ (Fig. 1, *right*), not $L$ as suggested in [4] (Fig. 5, *middle*), consistent with recent results from [12].

Finally, amid intense competition to advance LLM performance, a key question is: which $N$, $D$, and $B$ yield the best trade-off between training speed and compute cost? Using our fit $B_{\text{crit}}$ law, we derive Pareto-optimal solutions to these competing objectives, and show that small, overtrained models can be best—offering both faster steps and greater parallelism via larger $D$ (and thus higher $B_{\text{crit}}$).

Key findings and takeaways are highlighted in the paper. Our main contributions are:

- The first large-scale empirical study varying weight decay $\lambda$ across $N$, $D$, and $B$ in LLMs.
- Showing the AdamW timescale obeys a power law, enabling $\lambda_{\text{opt}}$ for any $N$, $D$, $B$ (Sec. 2).
- A new method for estimating $B_{\text{crit}}$, suitable for any LR schedule or optimizer (Sec. 3.2).
- Confirmation that both $B_{\text{opt}}$ and $B_{\text{crit}}$ scale as power laws in $D$ (Sec. 3).
- New methods for selecting $N$, $D$, and $B$ to trade-off training time vs. compute (Sec. 4).

## 2 Scaling of the AdamW timescale $\tau$, and optimal weight decay $\lambda_{\text{opt}}$

### 2.1 Background: $\mu$P, AdamW, and $\tau_{\text{epoch}}$

**$\mu$P**   $\mu$P is increasingly used in LLM training [13–18]. With $\mu$P, base HPs are tuned on a *proxy* model and then transferred to wider [5] and deeper [19] models. Given the width of the proxy model, $d_p$, and target, $d_t$, $\mu$P prescribes scaling factors to apply to the LR, initial weight variance, and other base HPs. In particular, the optimal base LR, $\tilde{\eta}_{\text{opt}}$ is scaled down to $\eta_{\text{opt}} = (d_p/d_t)\tilde{\eta}_{\text{opt}}$.

While $\mu$P enables the same base LR to be used across different $N$, $\tilde{\eta}_{\text{opt}}$ has empirically been found to *vary with $B$* [5, 20, 21, 16]. Moreover, recent work has also observed $\tilde{\eta}_{\text{opt}}$ *decreasing in $D$*, leading to proposals for scaling $\tilde{\eta}_{\text{opt}}$ as a (decreasing) power law in $D$ [16, 22].

**The EMA view of AdamW**   Rather than adjusting $\eta$ as $D$ scales, Wang and Aitchison [1] proposed that, if using the AdamW optimizer [23] with $\mu$P, then the weight decay, $\lambda$, should instead be adjusted. To see this, note an AdamW update at each step, $t$, can be expressed in terms of $\eta\lambda$ as:

$$\theta_t = (1 - \eta\lambda)\theta_{t-1} - \eta\frac{\hat{m}_t}{\sqrt{\hat{v}_t} + \epsilon} \tag{1}$$

Here, $\eta$ is the $\mu$P-adjusted LR, and $\hat{m}_t$ and $\hat{v}_t$ are (bias-corrected) exponentially-weighted moving averages (EMAs) of gradients and squared gradients [24]. Wang and Aitchison [1] observed AdamW's parameters, $\theta_t$, can *also* be viewed as an EMA—of weight *updates*. Specifically, the standard EMA form $y_t = (1 - \alpha)y_{t-1} + \alpha x_t$ matches AdamW when $y_t = \theta_t$, $\alpha = \eta\lambda$, and $x_t = -\frac{1}{\lambda}\frac{\hat{m}_t}{\sqrt{\hat{v}_t}+\epsilon}$. The quantity $1/\alpha = 1/\eta\lambda$ provides a measure of the number of iterations (i.e., steps) over which updates are averaged; [1] denotes it as $\tau_{\text{iter}}$. They show that if the timescale is measured in *epochs* as $\tau_{\text{epoch}} = \tau_{\text{iter}}/M$, where $M$ is iterations-per-epoch, then the optimal $\tau_{\text{epoch}}$ remains stable when $N$ or $D$ scale (on image tasks). I.e., if $M$ scales up, $\lambda$ should be scaled *down* to maintain constant $\tau_{\text{epoch}}$.

## 2.2 Methods: The AdamW timescale for LLMs, $\tau$, and its scaling

Since LLM pre-training only uses one "epoch" of data, we normalize the timescale as $\tau = \tau_{\text{iter}}/S$, where $S$ is the total number of optimization steps.[1] Moreover, since $S=D/B$,

$$\tau = \frac{B}{\eta\lambda D} \tag{2}$$

$\tau$ reflects the fraction of past iterations to include in the final weights. While Wang and Aitchison [1] did not vary $B$, their work suggests this fraction should remain constant as $B$ scales. We hypothesize that when moving from compute-efficient to overtrained LLMs, updates can be integrated over a smaller fraction of the data; specifically, that $\tau_{\text{opt}}$ decreases as a power law in TPP $:= D/N$:

$$\tau_{\text{opt}}(\text{TPP}) = c_\tau \cdot \text{TPP}^{m_\tau} \tag{3}$$

where $c_\tau$ and $m_\tau$ are parameters to be fit. Appendix Algorithm 1 summarizes the fitting procedure. Taking the $\mu$P-adjusted $\eta$ as our LR, $\lambda_{\text{opt}}$ can be computed from Eqs. (2) and (3):

$$\lambda_{\text{opt}} = \frac{B}{\eta \cdot D \cdot \tau_{\text{opt}}(\text{TPP})} = \frac{B \cdot \text{TPP}^{-m_\tau}}{c_\tau \cdot \eta \cdot D} \tag{4}$$

## 2.3 Experimental details

We use a GPT2-like LLM [25], with ALiBi embeddings [26] and SwiGLU [27]. We train on SlimPajama [28] and always evaluate over a held-out set of 1.1B tokens. We use AdamW and $\mu$P, with $\mu$P HPs derived from a smaller proxy model, and a linear LR schedule, with a 10% warmup followed by decay-to-zero [29]. Appendix C has full experimental details.

## 2.4 Results: $\tau$ and scaling $\lambda$

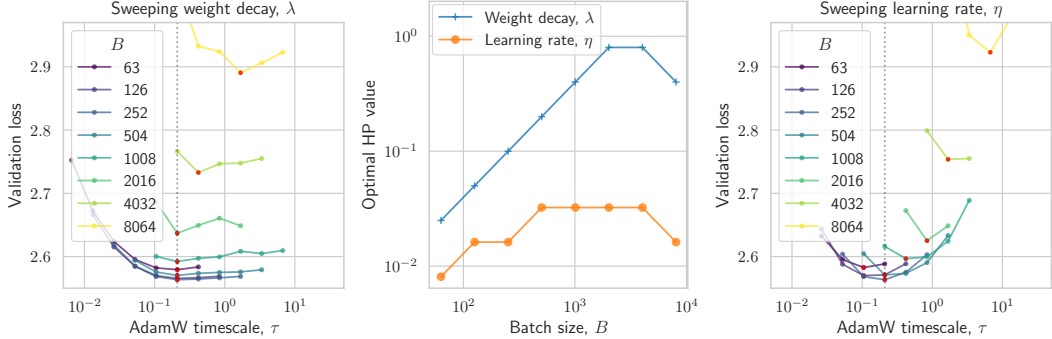

Figure 2: (610M 20TPP): For each $B$, we sweep $\lambda$ and find $\tau_{\text{opt}}$ (*left*). $\tau_{\text{opt}}$ is stable around 0.21 for $B \in [63, 2016]$, meaning $\lambda_{\text{opt}}$ scales linearly with $B$ over this range (*middle*). When sweeping $\eta$ (*right*), the lower boundary over all curves is a bowl with a minimum at 0.21; the smallest $B$ settings have $\tau_{\text{opt}}$ within $2\times$ of this value, but as $B$ increases, $\tau_{\text{opt}}$ quickly drifts higher.

---

[1]Because it depends on $\eta$, timescale $\tau$ *varies* during LR decay. Here we compute $\tau$ at peak LR. Notably, if two training runs share the same LR schedule (shape) and $\tau$ (at peak LR), their final AdamW timescale (EMA contributions over the *data*) will match—even if $B$ differs. Appendix E.4 provides further details.

Table 1: (610M, 20TPP) Validation losses comparing tuning $\lambda$ vs. $\eta$ across $B$ (data from Fig. 2).

| $\tilde{\eta}$ | $\lambda$ | $B=$ 63 | 126 | 252 | 504 | 1008 | 2016 | 4032 | 8064 |
|---|---|---|---|---|---|---|---|---|---|
| 1.6e-02 | 0.1 | 2.595 | 2.570 | **2.563** | 2.573 | 2.599 | 2.649 | 2.755 | 2.923 |
| Tuned | 0.1 | 2.583 | 2.570 | **2.563** | 2.571 | 2.597 | **2.625** | 2.754 | 2.923 |
| 1.6e-02 | Tuned | **2.579** | **2.565** | **2.563** | **2.570** | **2.592** | 2.637 | **2.733** | **2.891** |

> **Finding 1**: *The optimal $\tau$ remains stable as $B$ scales; $\lambda_{opt}$ scales linearly with $B$ (Fig. 2).*

As $B$ increases and $\lambda$ is tuned, we find that $\tau_{\text{opt}}$ remains roughly constant—i.e., changes in $B$ lead to commensurate changes in $\lambda_{\text{opt}}$ (Fig. 2, *middle*)—but only up to a certain point, after which $\tau_{\text{opt}}$ begins to drift. The drift point corresponds to the critical batch size $B_{\text{crit}}$, above which gradient information no longer scales linearly with $B$ and diminishing returns set in (Sec. 3).[2]

If $\eta$ is tuned instead, $\eta_{\text{opt}}$ fails to scale with $B$ up to $B_{\text{crit}}$ (Fig. 2, *middle*); we observe instead a certain maximum $\eta$, above which training becomes unstable; above this, loss spikes occur from which training does not recover. Consequently, $\eta$ has less flexibility to scale with $B$; training stability is more fundamental than timescale.

In general, LLMs typically train faster and utilize hardware better with larger $B$, but only up to $B_{\text{crit}}$: beyond this point, much more data (and compute) is needed to obtain the same loss, without meaningfully reducing the total number of sequential training steps (the poor trade-off for $B > B_{\text{crit}}$ is depicted in Fig. 4). Furthermore, Sec. 3 will show that there is also an *optimal* batch size, $B_{\text{opt}}$, below which loss is worse and utilization/parallelism suffer (because batches are small). In practice, LLMs should therefore be trained in the regime $B_{\text{opt}} \leq B \leq B_{\text{crit}}$. Notably, this is precisely the range where we have shown weight decay to scale predictably with batch size; our findings therefore support the direct optimization of weight decay in the most practically relevant training regimes.

> **Finding 2**: *With AdamW, we should adjust $\lambda$, not $\eta$, as $B$ changes.*

Since tuning at scale is infeasible, we need a recipe for selecting HPs in advance. Unlike $\eta$, optimal $\lambda$ follows a predictable relationship with $B$ (Fig. 2, *middle*), making $\lambda$ the more *viable* target for real-world adjustment. Moreover, since $\eta$ has less flexibility to maintain optimal timescale, we hypothesize adjusting $\lambda$ could also be more *effective*. We tested this by comparing either tuning $\eta$ (using a default $\lambda=0.1$—standard practice in LLM pre-training [3, 32–34])—or tuning $\lambda$ (using the $\mu$P proxy-tuned $\eta$). Tuning $\lambda$ was strictly superior in 6 of 8 cases (Table 1).

We also compared adjusting $\lambda$ versus $\eta$ as $D$ changes. For a 111M 200TPP model, default HPs obtain a loss of 2.810, tuning $\eta$ achieves 2.808, and tuning $\lambda$ obtains 2.805. While differences are small, the key point is that, when scaling $B$ or $D$: optimizing $\lambda$ alone is viable and effective.

> **Finding 3**: *$\tau_{opt}$ decreases as a power law in TPP; the law holds at scale (Fig. 1, left).*

At each $N$ and $D$, we calculated $\tau_{\text{opt}}$ over all $(\lambda, B)$ pairs; we then fit Eq. (3) to the results. Full details are in Appendix E.2. A precise power law emerges ($R^2=0.975$), with an optimal $\tau$ around 1.0 at 1 TPP, decreasing to 0.01 at 1000 TPP. 10th and 90th percentiles of fitted $m_\tau$ over all points are (-0.529, -0.507) (computed as in [3] by bootstrapping: re-fitting on 80% of points, 1000$\times$), indicating a reliable power-law trend. A decreasing $\tau$ stands in contrast to the prescription of Wang and Aitchison [1] (for multi-epoch training), who advocated keeping $\tau$ constant as $D$ changes.

Fig. 1 (*left*) includes four (labeled) points not used in fitting, but computed later to evaluate predictive ability. Even though some of these points are *interpolated* in terms of the fitting range (i.e., TPP range), they nevertheless represent much greater scales; e.g., training a 3.3B-30TPP model requires 1000$\times$ the FLOPs compared to training the 111M-20TPP model (whose data point is plotted nearby). In other words, the law generalizes across at least 3 orders of magnitude in compute.

---

[2]Training with $B > B_{\text{crit}}$ can be viewed as training with less *effective* data, thus decreasing the TPP, and shifting $\tau_{\text{opt}}$ higher (by our power law) and thus $\lambda_{\text{opt}}$ *lower*. A decrease in the optimal learning rate $\eta_{\text{opt}}$ when $B > B_{\text{crit}}$ has been previously observed [30, 31]. These works measure $\eta_{\text{opt}}$ at very large batch sizes, e.g., 10$\times$ or 100$\times$ $B_{\text{crit}}$, which was not practical at the scales that we trained. Nevertheless, our observation of a similar "surge" phenomenon with $\lambda_{\text{opt}}$, occurring in AdamW rather than vanilla Adam, motivates further study.

**Discussion** The $\tau$ scaling law also predicts previously-observed HP scaling in the literature. E.g., for a fixed $N$, Eqs. (2) and (3) together imply: $\eta_{\text{opt}} = B \cdot c_{\eta_D} \cdot D^{m_{\eta_D}}$ , where $c_{\eta_D}$ and $m_{\eta_D}$ are parameters. This matches Equation (1) in [16], and our implied $m_{\eta_D}$ is close to their fit value (see Appendix E.3.1). [22] also scales $\eta$ as a power law in $D$. They note that fitted power law exponents are similar when $B$ is doubled, although the optimal $\eta$ is "higher". More precisely, we can see from their Figure 13 that the optimal $\eta$ appears to, in fact, also double—consistent with the derived $\eta_{\text{opt}}$ equation above.

> ***Key takeaway 1**: With AdamW, you can find $\tau_{opt}$ for a small $N, D$ by tuning $\lambda$. From there, $\tau_{opt}$ scales $\propto (D/N)^{-0.5}$. At larger $N, D$, set $\lambda_{opt}$ via Eq. (4) and enjoy well-tuned models.*

## 3  Scaling of optimal batch size $B_{\text{opt}}$ and critical batch size $B_{\text{crit}}$

We now develop methodology to estimate $B_{\text{opt}}$ and $B_{\text{crit}}$ over the dimensions of total training tokens $D$, total compute FLOPs $C$, and validation loss $L$. Results show power-law scaling of both $B_{\text{opt}}$ and $B_{\text{crit}}$ with $D$, enabling estimation of $B_{\text{crit}}$ at scale via a small number of test runs at modest budgets.

### 3.1  Background: $B_{\text{opt}}$ and $B_{\text{crit}}$

$B_{\text{opt}}$   As noted above, recent work has pursued an *optimal* $B$: the $B$ achieving lowest loss given $N, D$. Hu et al. [17] fit $B_{\text{opt}}$ using a power law in (estimated) loss, and use $\mu$P to set $\eta$. Joint power laws for optimal $\eta$ and $B$ have also been fit [7, 35, 36]. E.g., [7] estimated $B_{\text{opt}} = 0.292C^{0.3271}$ (in tokens); we refer to this fit as $B_{\text{deepseek}}$ below. Li et al. [37] found $\eta_{\text{opt}}$ to scale in $N, D$, while $B_{\text{opt}}$ primarily scales in $D$. Qwen2.5 [38] also report studying how $\eta_{\text{opt}}$ and $B_{\text{opt}}$ scale with $N$ and $D$ (across dense and mixture-of-expert LLMs), but without further details.

$B_{\text{crit}}$   Let $D$ be the number of training tokens required to reach target loss $\hat{L}$ when using a batch size of $B$. $S = D/B$ is the corresponding number of optimization steps. Doubling a small $B$ doubles per-step gradient information; $\hat{L}$ can be reached in half the optimization steps, using the same total $D$ (so-called "perfect scaling" [39, 40]). But as $B$ increases further, step-wise gradient information becomes more and more redundant: eventually, much larger $D$ is required to reach $\hat{L}$, and $S$ decreases only marginally. McCandlish et al. [11] show that $\langle D, S \rangle$ pairs can be well fit by the equation:

$$S/S_{\min} - 1 = (D/D_{\min} - 1)^{-1} \tag{5}$$

where $S_{\min}$ and $D_{\min}$ are parameters to be fit. Intuitively, $D_{\min} < D$ is the asymptotically minimum number of tokens that can reach $\hat{L}$ (achieved with $B_{\text{opt}}$) and $S_{\min} < S$ is the asymptotically minimum number of *steps* that can reach $\hat{L}$ (achieved as $B \to \infty$). Eq. (5) defines a hyperbolic curve, like those in Fig. 4, where $B$ controls the position on the curve and can be set depending on the importance of time (higher $B \to$ higher $D$, lower $S$) or compute (lower $B \to$ lower $D$, higher $S$).

**Definition 3.1.** The *critical batch size* at $\hat{L}$ is defined from the fit of Eq. (5) as $B_{\text{crit}} = D_{\min}/S_{\min}$.

From Eq. (5), we can derive (Appendix F.1), for a given $B$, the $D$ needed compared to $D_{\min}$:

$$D = D_{\min}(1 + B/B_{\text{crit}}) \tag{6}$$

Eq. (6) implies that when $B = B_{\text{crit}}$, we require $2 \times D_{\min}$ tokens (and $2 \times S_{\min}$ steps) to reach $\hat{L}$. $B_{\text{crit}}$ is a *transition* point along the $D$ vs. $S$ curve: for $B > B_{\text{crit}}$, much higher $D$ is needed for only small reductions in $S$ (Fig. 4). Kaplan et al. [4] refer to $B_{\text{crit}}$ as the *optimal compromise* between time and compute. They determine $B_{\text{crit}}$ at smaller scales and fit a power law for $B_{\text{crit}}$ as a function of $\hat{L}$.

Eqs. (5) and (6) also imply $B_{\text{opt}}$ is theoretically equal to 1. In practice, loss degrades below a particular $B_{\text{opt}}$ [17, 7, 35], a finding that "appears to contradict the conventional wisdom" about $B_{\text{crit}}$ [35]. With well-tuned $\lambda$, we find small differences in loss across small $B$, suggesting Eq. (5) may nevertheless provide a good fit to observed data. Appendix B has further discussion.

In recent work, Zhang et al. [12] define $B_{\text{crit}}$ as the point where $D = 1.2 \times D_{\min}$. [12] uses a different training setup, with a constant LR, weight averaging, and no weight decay. Notably, they observe little change in $B_{\text{crit}}$ as $N$ varies at fixed $D$, but, for a 302M model, find power-law scaling in $D$ as $B_{\text{crit}} = 22.91D^{0.47}$ (in tokens), consistent with observed scaling across models at fixed TPP. See Appendix D.3 for further differences with Zhang et al. [12].

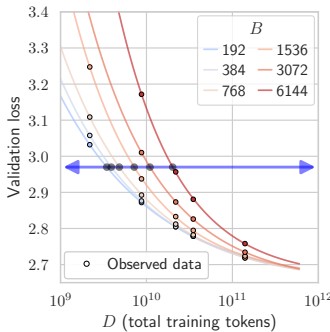

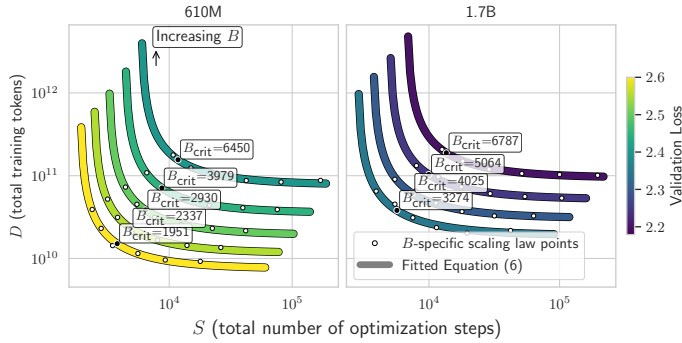

Figure 3: (111M): **Fitted $B$-specific power laws**, $L_B(D)$, for inferring steps to reach target loss $\hat{L}$ (arrowed blue line).

Figure 4: **Eq. (5) fits observed data well**: Increasing $B$ (moving leftward along curves) decreases optimization steps (x-axis), but requires more tokens (y-axis) to reach target loss (in color). $B_{\text{crit}}$ is the transition point along each fitted curve.

## 3.2 Methods: estimating $B_{\text{opt}}$ and $B_{\text{crit}}$, and their scaling

**Estimating $B_{\text{opt}}$**  We use the same experimental settings as Sec. 2.3. To ensure good HPs, we sweep $\lambda$ by factors of $2\times$ at each $B$, $D$, $N$, except at the largest scales (see Appendix Table 3) where we set $\lambda$ via the projected value from Eq. (4). In all figures, $B$ is reported in units of *sequences*.

**Estimating $B_{\text{crit}}$**  Unlike $B_{\text{opt}}$, measuring $B_{\text{crit}}$ requires training models with different $B$ *to the same* $\hat{L}$. Unfortunately, we do not know *a priori* how many steps are required to reach $\hat{L}$, yet we need this information to configure a LR schedule that reaches its minimum value on the final step (the typical setup, shown to be consequential in prior work [3, 41]). Unfortunately, it is not feasible to *search* for the precise steps needed, i.e., by conducting training runs with different schedules/step budgets.

McCandlish et al. [11] address this issue by performing a single training run at a constant LR, while Zhang et al. [12] also use a constant LR, but use weight averaging to frequently generate higher-quality checkpoints for evaluation (Appendix D.3).

In contrast, we desired a method agnostic to the LR schedule. We achieved this by fitting batch-size-specific power laws that model how *loss* scales with $D$. These laws allow us to accurately *interpolate* the $D$ required to reach $\hat{L}$. Fig. 3 depicts, for different $B$ and a given $\hat{L}$, the interpolated $D$ values (intersection points of arrowed line and fitted loss curves). The full process to obtain $B_{\text{crit}}$ at $\hat{L}$ is:

1. For each $B$, train over different $D$, and subsequently fit a $B$-specific power law $L_B(D) = E_N + D_{\text{const}} D^{-\beta}$ on the resulting loss values (fitted curves in Fig. 3).

2. Use fitted $L_B(D)$ to infer the $D_B$ needed to reach $\hat{L}$ as: $D_B = L_B^{-1}(\hat{L}) = \left( D_{\text{const}} / \hat{L} - E_N \right)^{\frac{1}{\beta}}$.

3. Fit Eq. (5) on the resulting $\langle D_B, S{=}D/B \rangle$ pairs, and obtain $B_{\text{crit}} = D_{\text{min}}/S_{\text{min}}$.

This method makes no assumptions about the LR schedule or optimizer, while enabling measurement of $B_{\text{crit}}$ at arbitrary losses without re-training. Appendix F.2 provides further details, including fits of $L_B(D)$ at other model scales (Fig. 9) and a summary of the full procedure (Algorithm 2).

**$B_{\text{opt}}$ and $B_{\text{crit}}$ scaling**  We collect $B_{\text{opt}}$ across different $N$ and $D$, and fit a power law in both data $D$ and compute $C$ (via the standard approximation $C \approx 6ND$ [4, 3]). For $B_{\text{crit}}$, we use the procedure described above to estimate $B_{\text{crit}}$ across multiple $\hat{L}$, across different $N$. From each $B_{\text{crit}}$ estimate, we obtain a pair $\langle D_{\text{min}}, B_{\text{crit}} \rangle$. We propose that $B_{\text{crit}}$ follows a power law in $D_{\text{min}}$, according to:

$$B_{\text{crit}}(D_{\text{min}}) = c_{B_{\text{crit}}} \cdot D_{\text{min}}^{m_{B_{\text{crit}}}} \tag{7}$$

Where $c_{B_{\text{crit}}}$ and $m_{B_{\text{crit}}}$ are fit on the $\langle D_{\text{min}}, B_{\text{crit}} \rangle$ pairs.

## 3.3 Results: $B_{\text{opt}}$ and $B_{\text{crit}}$

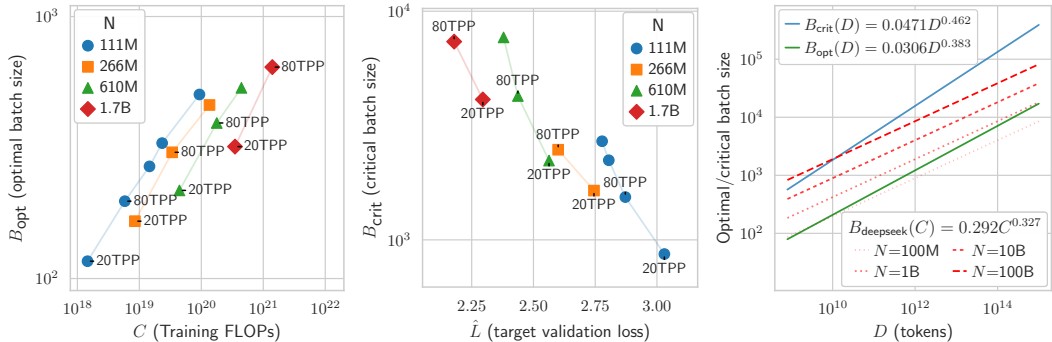

Figure 5: Prior work suggests $B_{\text{opt}}$ scales in $C$ (*left*) and $B_{\text{crit}}$ in loss (*middle*), but *this only holds at a fixed N/TPP* (same data as Fig. 1 *middle/right*); Fig. 1 shows scaling in $D$ is the fundamental relationship. Plotting $B_{\text{deepseek}}(C)$ law from [7] (*right*), but over $D$ (using $C \approx 6ND$ to obtain $C$ for the spurious dependence on $N$), we see [7] used generally efficient $B$ values (i.e., within the $B_{\text{opt}} < B < B_{\text{crit}}$ regime) despite fitting $C$ rather than $D$ ($B_{\text{opt}}$ and $B_{\text{crit}}$ lines from Fig. 1).

> **Finding 4**: *Eq. (5) provides a decent fit to the trade-off between training time and compute.*

Across different model scales and loss targets, we consistently find that our $\langle D, S \rangle$ pairs fit Eq. (5) well (examples in Fig. 4, appendix Fig. 10). Fits are worse at very small $B$, as noted above: smaller batches are not monotonically more efficient; Appendix B discusses some potential reasons for this.

> **Finding 5**: *$B_{\text{opt}}$ and $B_{\text{crit}}$ obey power laws in $D$ and $D_{\text{min}}$, not in $C$ or $L$.*

$B_{\text{opt}}$ and $B_{\text{crit}}$ datapoints fit power laws quite well (Fig. 1, *middle*, $R^2$=0.984) and (Fig. 1, *right*, $R^2$=0.940). 10th and 90th percentiles over all points are (0.367, 0.391) for fitted $m_{B_{\text{opt}}}$ and (0.491, 0.526) for $m_{B_{\text{crit}}}$ (computed as in Sec. 2.4). Note $m_{B_{\text{crit}}}$ is higher when fitted over *all* points (as opposed to only small-scale runs), partly reflecting the 111M points trending lower as TPP increases.

Our fitted $B_{\text{crit}}$ power law exponent is very close to that from [12]: 0.47 vs. 0.462. Given the many differences in approach (including dataset, use of weight decay, LR schedule, etc., Appendix D.3), this agreement suggests the fundamental relationship of $B_{\text{crit}}$ with $D$ persists across such differences.

Fig. 5 (*left*) plots $B_{\text{opt}}$ versus $C$ and Fig. 5 (*middle*) gives $B_{\text{crit}}$ versus $L$ using the same data as in Fig. 1. In each case, a power law does not fit all points (as proposed previously), but points at the same $N$, or same TPP, can roughly be linked by (parallel) lines. This is a consequence of power-law scaling in $D$ (see Appendix F.4). That is, scaling in $D$ is the fundamental scaling relationship: $B_{\text{opt}}$ and $B_{\text{crit}}$ both scale in $D$ regardless of TPP, model size, or loss—it is only when using another (misleading) scaling factor such as $C$ or $\hat{L}$ that TPP or model size appears important, as in these plots.

Fig. 5 (*right*) compares the recommended batch sizes from $B_{\text{deepseek}}$ to those from $B_{\text{opt}}$ and $B_{\text{crit}}$. Since $B_{\text{deepseek}}$ scales in $C$, it is larger for larger $N$. Over a range of modern model sizes, $B_{\text{deepseek}}$ values generally fall between our projected $B_{\text{opt}}$ and $B_{\text{crit}}$, varying in the extent to which they are compute-efficient (close to $B_{\text{opt}}$) or time-efficient (close to $B_{\text{crit}}$).

> **Finding 6**: *Weight decay affects the accuracy of fitted batch size scaling laws.*

Prior work has typically held $\lambda$ fixed when fitting batch-size scaling laws [7, 35, 12]. Doing so not only degrades loss (Sec. 2) but also reduces the accuracy and generality of the fitted scaling relationships. We demonstrate this in appendix Table 5: rather than tuning $\lambda$ for each $B$, we train with several fixed $\lambda$ values across all runs. As Table 5 shows, increasing $\lambda$ systematically raises the estimated $B_{\text{opt}}$. This arises because the fundamental scaling variable is the AdamW timescale $\tau = B/(\eta\lambda D)$: when $\lambda$ increases, the batch size that minimizes loss must increase proportionally to preserve the optimal $\tau$. In Appendix F.5, we show that these effects distort the fitted power-law slope and reduce fit quality ($R^2$), leading to scaling laws that do not generalize to large-scale training—even

if the same fixed weight decay is used there. When $\lambda$ is tuned to maintain the optimal timescale (final row of Table 5), the resulting $B_{\text{opt}}$ follows a clean and accurate power law.

Similar distortions occur for $B_{\text{crit}}$ when $\lambda$ is fixed (Appendix F.5).

> **Key takeaway 2**: *You can estimate $B_{opt}$ and $B_{crit}$ for a small $N$ by training with different $B$, $D$ and $\lambda_{opt}$, and computing loss. From there, $B_{opt} \propto D^{0.4}$ and $B_{crit} \propto D^{0.5}$. At larger $N$ and $D$, Eq. (6) lets you estimate trade-offs in FLOPs ($\propto D$) vs. training time ($\propto S = D/B$) at different $B$.*

## 4 Training settings for balancing time and compute

### 4.1 Background: compute-optimal and overtrained models

Given a fixed training FLOPs budget, $C$, how should we allocate model size $N$ versus number of training tokens $D$ in order to minimize *loss*? Hoffmann et al. [3] propose to model loss as:

$$L(N, D) = E + N_{\text{const}}N^{-\alpha} + D_{\text{const}}D^{-\beta} \tag{8}$$

$N_{\text{const}}$, $\alpha$, $D_{\text{const}}$, and $\beta$ are parameters fit on observed training runs. From Eq. (8), [3] derives functions for loss-optimal $N_{\text{opt}}(C)$ and $D_{\text{opt}}(C)$ (constraining $L(N, D)$ by $C \approx 6ND$). Results indicate $N_{\text{opt}}$ and $D_{\text{opt}}$ scale roughly equally as $C$ increases, with the optimal $D/N$ ratio relatively constant at around 20 TPP. Replication studies have found similar results [9, 35], and 20 TPP has become a rule-of-thumb for compute-optimal training [13, 12].

Overtrained, inference-efficient models [10, 42, 43] have largely trained with similar batch sizes to those used in compute-optimal training; such efforts should now consider training with much greater data parallelism, leveraging our finding that $B_{\text{opt}}$ and $B_{\text{crit}}$ will be higher given the higher training $D$.

### 4.2 Methods: exploring the trade-offs of FLOPs vs. time

To compare models of different sizes on a common temporal axis, we must map number-of-optimization-steps to a common temporal scale. Our initial approximation is Training Time $\propto \text{Total FLOPs}/B$, which is also FLOPs *per token* times number of steps. E.g., if FLOPs $\approx 6ND$, Training Time $\approx 6ND/B = 6N \cdot S$. This aligns well with our measured runtimes: doubling N doubles step time; doubling B halves wall-clock time (for the same $S$).

Now, assume a model of size $N$ can train to loss $\hat{L}$ using $D_{\text{min}}$ tokens (here *min* denotes using $B_{\text{opt}}$). Let us refer to $N$ and $D_{\text{min}}$ as a *base setting*. A variety of $N$, $D_{\text{min}}$ pairs can reach $\hat{L}$ in the $B_{\text{opt}}$ setting, from small models trained on many tokens, to large models trained on fewer tokens. [3] refers to these as iso-loss *contours* of Eq. (8). Suppose a given base setting requires $C(N, D_{\text{min}})$ FLOPs. From this setting, we may increase $B$ to decrease training time (fewer steps), but Eq. (6) indicates a need for $(1 + B/B_{\text{crit}})$ extra *data* in order to reach the same $\hat{L}$. If FLOPs is linear in $D$ (as in $C = 6ND$), we will require the same proportion of extra FLOPs, i.e.,

$$C_+(N, D_{\text{min}}, B) = C(N, D_{\text{min}})(1 + B/B_{\text{crit}}(D_{\text{min}})) \tag{9}$$

where $C_+(N, D_{\text{min}}, B)$ denotes the total FLOPs needed at $B > B_{\text{opt}}$, and $B_{\text{crit}}(D_{\text{min}})$ captures that the excess FLOPs depends on $B_{\text{crit}}$, which itself scales with $D_{\text{min}}$. In other words, the base setting dictates $B_{\text{crit}}$, and $B/B_{\text{crit}}$ dictates the excess FLOPs.

Consider a target FLOP budget of $C_+(N, D_{\text{min}}, B) = \hat{C}$ and the goal of reaching $\hat{L}$ *as fast as possible*. Since time $\propto \text{Total FLOPs}/B$, time is minimized by maximizing $B$. However, by construction, $B$ is not a free variable: it is constrained by Eq. (9) and can be expressed as a function of $N$ and $D_{\text{min}}$:

$$B(N, D_{\text{min}}) = \left(\frac{\hat{C}}{C(N, D_{\text{min}})} - 1\right) B_{\text{crit}}(D_{\text{min}}) \tag{10}$$

Time is therefore minimized by finding $N$, $D_{\text{min}}$ that maximize this function (over all the $N$, $D_{\text{min}}$ that train to loss $\hat{L}$). The $\hat{C}/C$ ratio is a measure of the *excess* FLOPs that can be spent toward increasing $B$; it is largest when $C(N, D_{\text{min}})$ is smallest, i.e., when $N$ and $D_{\text{min}}$ is most compute-efficient (i.e., $N/D_{\text{min}} \approx 20$ TPP). But Eq. (10) as a whole captures an elegant tension between compute efficiency

and $B_{\mathrm{crit}}$: we can maximize $B$ (and minimize training time) by either (1) minimizing the FLOPs of the base setting (generating more excess FLOPs for increasing $B$), or (2) maximizing $D_{\min}$ (overtraining, which increases $B_{\mathrm{crit}}(D_{\min})$). For a given $\hat{C}$, either (1) or (2) may take precedence.

We use the following procedure to explore the time vs. compute Pareto frontier for a target loss $\hat{L}$:

1. Fit Eq. (8) on our $B_{\mathrm{opt}}$ training runs. Express resulting $L(N, D_{\min})$ as $D_{\min\,\hat{L}}(N)$.

2. Using $D_{\min\,\hat{L}}(N)$, get contour points $\langle N, D_{\min}\rangle$ of the given $\hat{L}$. Each such pair consumes $C(N, D_{\min}) \approx 6ND_{\min}$ FLOPs and takes $C(N, D_{\min})/B$ time (rightmost points on Fig. 6 curves).

3. Use Eq. (9) to compute $C_+(N, D_{\min}, B)$ as we scale $B$ (crucially, using the estimate of $B_{\mathrm{crit}}$ from fitted Eq. (7)), and generate further points along each curve.

4. The non-dominated points over all curves provide the time vs. compute Pareto frontier.

## 4.3    Results: balancing time and compute

We carry out this procedure using model sizes of 150M, 210M, 550M, 1.1B, and 2.1B, and a loss target of $\hat{L}$=2.6, yielding iso-loss contour points from 150M 600TPP to 2.1B 2TPP. Our fit of Eq. (8) yielded $\alpha$=0.313 $\approx \beta$=0.282, giving an optimal TPP ratio of $\approx$20.6 at $\hat{L}$=2.6.

> **Finding 7**: *Overtrained, but not undertrained, models are on the FLOPs vs. time Pareto frontier.*

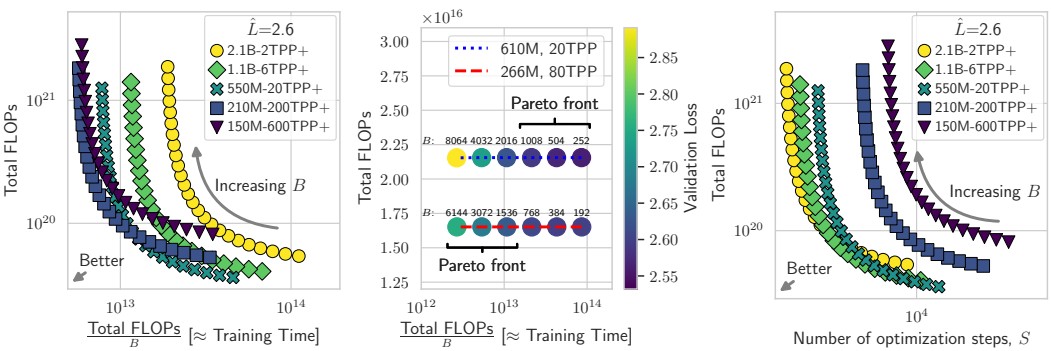

Figure 6: (*left*): Iso-loss curves illustrating time–compute Pareto frontier ($\hat{L}$=2.6). As $B$ increases along curves, more compute (y-axis), but less time (x-axis) is required. Here time $\propto$ Total FLOPs/$B$. (*middle*): Observed runs where some overtrained models (red line) are on frontier: $L$ in color, $B$ labeled. (*right*): Iso-loss curves, but where time = steps; a very different frontier emerges.

Specifically, when using Training Time $\propto$ Total FLOPs/$B$, we find overtrained models are FLOP-optimal at certain time budgets (Fig. 6, *left*). Compute-efficient 20 TPP are optimal in pure FLOPs (i.e., ignoring time), as expected, while compute-efficient and overtrained models dominate undertrained ($< 20$ TPP) models in time and FLOPs. Indeed, this is expected from Eq. (10): *under*training reduces both the excess FLOPs and $B_{\mathrm{crit}}$ terms, and thus is never optimal with this model of time.

> **Finding 8**: *When using $B \gg B_{opt}$, it is Pareto-inefficient to train to 20 TPP.*

Notice that Fig. 6 adds "…*TPP+*" to curve labels. Here the $+$ sign is a reminder that as we increase $B$, we require $(1 + B/B_{\mathrm{crit}})$ *extra data* to reach the same $\hat{L}$; i.e., points with higher $B$ are trained to a higher *actual TPP* than the base setting. For example, once the 2TPP+ curve in Fig. 6 reaches $10\times$ its minimum FLOPs, it is *actually* training at 20 TPP. Since starting from an undertrained base setting is never Pareto optimal (as just discussed above), *it is **always** suboptimal to train a model with a large $B$ to 20 actual TPP*. If large-batch training is needed, the configuration should start from a 20 TPP+ base setting and scale $B$ from there (to >20 TPP).

We can see this finding play out in real training runs. Fig. 6 (middle) demonstrates observed runs where our 266M 80TPP models *dominate* our 610M 20TPP models (i.e., in FLOPs *and* time)—when both train with large $B$. (Note in this plot, results are not iso-loss: the frontier is over $L$, $C$, and time.)

> **_Finding 9_**: _The Pareto-optimal settings depend on the formulation of time/parallelism strategy._

While $^{\text{FLOPs}}/_B$ is a good model of data parallel training, it does not incorporate the potential for model parallelism [44]. In the extreme we could assume that all $6N$ FLOPs could be executed concurrently per input token. Under this formulation, training time is proportional only to the number of steps, regardless of model scale. Fig. 6 (*right*) shows the Pareto frontier that would result from this formulation; if we pay no time cost for larger models, we can train faster by using undertrained large models, although, exactly as with overtrained models, they suffer in FLOPs.

While LLMs cannot be fully parallelized due to the inherent sequential nature of a Transformer's layer-by-layer computation, this formulation could be refined by incorporating depth or other architectural features. For example, Inbar and Sernau [45] predict training time via linear regression over total FLOPs and memory-copy operations, fit to real (single-TPU) runs. As formulations improve, different Pareto-optimal configurations will emerge.

> **_Finding 10_**: _Inaccurate $B_{crit}$ scaling leads to inaccurate Pareto-optimal configurations._

Because the Pareto frontier in Sec. 4 depends directly on the $B_{\text{crit}}$ power-law fit, any error in that fit produces corresponding errors in the predicted trade-offs between training time and compute. Accurate $B_{\text{crit}}$ estimation, in turn, depends on effective $\lambda$ tuning (Finding 6 and Appendix F.5). When $\lambda$ is fixed as in standard practice (and thus the $B_{\text{crit}}(D)$ slope is misestimated), $B_{\text{crit}}$ will be systematically over- or underpredicted at scale, altering the computed frontier and the apparent Pareto-optimal configurations. To illustrate, artificially varying the $B_{\text{crit}}$ exponent changes which models appear on the frontier: as the exponent increases (and $B_{\text{crit}}$ rises), higher-TPP models move to the frontier; when $B_{\text{crit}}$ is underestimated, only low-TPP (e.g., 20 TPP) models appear Pareto-optimal. Hence, an inaccurate $B_{\text{crit}}$ scaling law produces misleading frontiers and can lead to unexpectedly-longer training durations and suboptimal compute allocations.

Recent work suggests that other estimators of $B_{\text{crit}}$ (such as those based on the gradient noise scale [11]) can also be systematically biased (Appendix D.2), leading to similar distortions in the Pareto frontier.

> **_Key takeaway 3_**: _To balance time and compute at a target loss, select $(N, D_{\min})$ from Eq. (8), determine $B_{crit}(D_{\min})$ via Eq. (7), and use Eq. (9) to estimate compute for any B. Under Time $\propto$ FLOPs$/B$, the resulting time–compute trade-off favors higher D (overtraining) (Fig. 6)._

## 5 Conclusion

We have presented a comprehensive empirical study of hyperparameter scaling laws in LLM pre-training, focusing on weight decay and batch size. Our approach leverages the AdamW timescale ($\tau$) to develop robust scaling relationships that predict optimal hyperparameter settings across a broad spectrum of model ($N$), dataset ($D$), and batch sizes ($B$). We demonstrated that optimal $\tau$ decreases as a power law with the tokens-per-parameter ratio, providing a systematic method to set weight decay optimally across diverse training scenarios.

Furthermore, we introduced a novel, practical methodology for estimating critical batch size ($B_{\text{crit}}$). Our findings diverge from influential prior work that tied $B_{\text{crit}}$ predominantly to compute or loss, while agreeing with the recent findings of [12] that underscore dataset size as the principal scaling factor. Additionally, we showed that contrary to previous studies suggesting optimal batch size ($B_{\text{opt}}$) scales primarily with compute, it also exhibits a clear power-law dependence on $D$.

Also, our analysis of Pareto-optimal configurations reveals an important strategic advantage for smaller, overtrained models in scenarios where rapid training and high parallelism are prioritized.

Appendix B notes limitations and directions for further study suggested by our results. In particular, as inference-time scaling comes to the fore, inference time and compute must also be considered as first-class Pareto objectives. Moreover, finer-grained configuration decisions, such as model depth and context length, should be considered along with $N$, $D$, and $B$.

## Acknowledgments and Disclosure of Funding

We thank the NeurIPS reviewers for their helpful feedback. None of the authors received third-party funding or third-party support for this work. None of the authors have financial relationships with outside parties that could potentially be perceived to influence this research.

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

# A Broader impacts

This paper presents methods to train LLMs more efficiently: practitioners can use our methods to reduce the total compute FLOPs used to train models, subject to time constraints. Given the intense pressure to advance LLM capabilities as quickly as possible, our methods can therefore reduce the associated environmental and financial costs of LLM training [46, 47].

Moreover, hyperparameter tuning is a key contributor to these costs, and impairs equity in AI research, as tuning success depends directly on researcher finances [48]. We hope our exploration of optimal hyperparameter scaling can reduce the burden of hyperparameter tuning at scale and thus improve equity in AI.

# B Limitations

While our findings corroborate prior work and provide strong evidence for the proposed scaling laws in $\tau$, $B_{\text{opt}}$, and $B_{\text{crit}}$, there are several limitations that merit further study.

**EMA perspective**    As the EMA perspective regards parameters $y_t$ as a function of updates $x_t$, it fails to account for $x_t$ actually depending on earlier values of $y_t$ (e.g., $y_{t-1}$). Yet although this perspective has formal limitations, we nevertheless find it a useful *conceptual* model of training, as it predicts behavior that is supported by experiments.

**Optimization and training setup**    Our work focuses on AdamW (the standard optimizer for LLM training). While the EMA perspective applies directly to other optimizers that use decoupled weight decay, such as Sophia [49] and MuonClip [50], it may not apply to approximate second order methods, e.g., Shampoo [51]. However, it can be used when applying AdamW (and related optimizers) in Shampoo's eigenbasis, which was shown to be effective in SOAP [52].

We present results with a single (standard) learning rate schedule. Our method for obtaining $B_{\text{crit}}$ estimates would be quite efficient with a warmup-stable-decay (WSD) schedule [17, 53], as we could perform a single training run with each batch size, but decay at various milestones in order to get points along the scaling law, essentially following the approach in Hägele et al. [41], but with separate laws for each batch size.

We used the maximal update parameterization in all experiments, which generates a learning rate $\eta$ adjustment for each model width. Our results suggest this approach enables good models at arbitrary $N$, $D$, and $B$ when combined with adjustments to $\lambda$. This strategy is informed by our experiments comparing re-adjusting $\eta$ vs. $\lambda$ in Sec. 2.4. However, it is not feasible, at this scale, to verify whether substantially better models could be obtained by sweeping the full cross-product of $\eta$ and $\lambda$ values.

Our study specifically focuses on the practically important setting of single-epoch LLM pre-training. Wang and Aitchison [1] indeed noted differences in optimal $\tau$ when using multi-epoch training, possibly due to data repetition. Reconciling these differences by isolating the effects of repetition versus scale is an interesting follow-up direction.

Here we only experimented with a single dataset, vocabulary, and context length. We obtained a similar $B_{\text{crit}}$ scaling law to Zhang et al. [12], but it would be interesting to see if differences in the coefficient of our power laws could be attributed to specific differences in approach (e.g., differences in dataset, context length, learning rate schedule, use of weight decay, etc.). Appendix D.3 has further discussion of differences with Zhang et al. [12].

We have also not explored how changes in numerical precision could affect scaling laws. Recent work [54] showed that, in terms of scaling laws, lower precision reduces the model's *effective* parameter count. This suggests precision would have no impact on scaling of $B_{\text{opt}}$ or $B_{\text{crit}}$, which do not scale in $N$. Lower precision, however, could increase the *effective* TPP (via smaller effective $N$), thereby altering $\tau_{\text{opt}}$.

**Small batches, large batches, and dynamic batch sizing**    We consistently find that smaller and smaller batches do not grow asymptotically closer to $D_{\text{min}}$, as predicted by theory, but eventually degrade in loss. One possibility is that $\lambda$ tuning is not sufficient with very small $B$, and further tuning of other hyperparameters may be needed, such as the Adam $\beta$ parameters (as suggested in recent

Table 2: Model architectures used in experiments

| Model | $d_{model}$ | $n_{layers}$ | $d_{head}$ |
|-------|-------------|--------------|------------|
| 111M  | 768         | 10           | 64         |
| 266M  | 768         | 32           | 64         |
| 610M  | 2048        | 10           | 64         |
| 1.7B  | 2048        | 32           | 64         |
| 3.3B  | 2048        | 64           | 64         |

Table 3: Models, tokens-per-parameter (TPP) and corresponding dataset sizes (in tokens) used in main experiments. We also list the total number of batch sizes, $B$, trained at each scale and TPP, as well as the number of $B$ for which we tuned $\lambda$. For $B$ where $\lambda$ was not tuned, it was inferred via the $\tau_{\text{opt}}$ scaling law (Sec. 2). Additional sweeps of $\eta$ were done at each $B$ at 610M-20TPP scale for the experiments in Sec. 2.4. Around 400 different LLMs were trained in total across all the experiments.

| Model | TPP  | $D$     | Number of $B$ | Number of B with $\lambda$ tuned |
|-------|------|---------|---------------|----------------------------------|
| 111M  | 20   | 2.19B   | 8             | 8                                |
| 111M  | 80   | 8.76B   | 8             | 8                                |
| 111M  | 200  | 21.9B   | 7             | 7                                |
| 111M  | 320  | 35.0B   | 8             | 8                                |
| 111M  | 1280 | 140.1B  | 6             | 1                                |
| 266M  | 20   | 5.31B   | 7             | 7                                |
| 266M  | 80   | 21.2B   | 7             | 7                                |
| 266M  | 320  | 85.0B   | 6             | 6                                |
| 266M  | 1280 | 339.8B  | 1             | 0                                |
| 610M  | 20   | 12.1B   | 8             | 8                                |
| 610M  | 80   | 48.5B   | 7             | 7                                |
| 610M  | 200  | 121.3B  | 6             | 6                                |
| 610M  | 320  | 194.1B  | 2             | 1                                |
| 1.7B  | 20   | 34.3B   | 7             | 7                                |
| 1.7B  | 80   | 137.2B  | 7             | 1                                |
| 1.7B  | 320  | 548.6B  | 1             | 0                                |
| 3.3B  | 20   | 66.5B   | 1             | 0                                |
| 3.3B  | 23   | 76.5B   | 1             | 0                                |
| 3.3B  | 30   | 99.8B   | 2             | 1                                |

work [35, 12, 55]). Some preliminary tests using the $\beta_2$ scaling rule from Marek et al. [55] showed loss improvements at small $B$. Since we are unlikely to train with small batches at scale, and using them even with smaller LLMs significantly impairs our ability to train both efficiently and quickly, it is unfortunately difficult to justify further exploration in this direction.

Regarding large batches, our methods do not account for the many practical systems-related issues, including bandwidth and communication overheads, memory limits of hardware, synchronization delays, etc. Moreover, as batch sizes increase, techniques such as optimizer sharding may be needed, which further complicate performance model [33]. Our scaling laws do, however, explicitly define a practically relevant regime of training batch sizes: $B_{\text{opt}} \leq B \leq B_{\text{crit}}$. Practitioners can leverage this identified regime alongside system-specific profiling (e.g., evaluating utilization at various batch sizes) to select optimal settings balancing algorithmic and systems constraints.

Exploring optimal dynamic batch sizing is a natural future direction for our work. While the potential gains were found to be small in theory by McCandlish et al. [11], more recent work has found significant wall clock speedups [56].

## C   Experimental Details

Table 2 provides details on the model architecture and hyperparameters for models used in the experiments. Table 3 provides, for each model scale and TPP, the dataset sizes used in training, the

Table 4: Tuned hyperparameters for $\mu$P proxy model

| | |
|---|---|
| $\sigma_{W,\text{base}}$ | 8.67e-02 |
| $\tilde{\eta}$ | 1.62e-02 |
| $\alpha_{\text{input}}$ | 9.17 |
| $\alpha_{\text{output}}$ | 1.095 |

number of batch sizes tested, and the number of batch sizes for which $\lambda$ was tuned. Around 400 models in total were trained for the main experiments.

All the models in our main experiments were trained on the SlimPajama dataset [28], a cleaned and deduplicated version of the RedPajama dataset. We use the GPT-2 [25] vocabulary of size 50257, and a context length of 2048 tokens. Following standard practice, we do not apply weight decay or bias to LayerNorm layers. AdamW settings are $\beta_1 = 0.9$, $\beta_2 = 0.95$, and $\epsilon = 1\mathrm{e}{-8}$. Validation loss is always computed over a held-out 1.1B tokens, regardless of training TPP. We report cross-entropy loss. By default we parameterize with $\mu$P, with hyperparameters set via proxy tuning, as described below.

For a given TPP, all models have the exact same warmup phase: a linear warmup of the learning rate from 0 to the maximum value. In all our runs, warmup was 10% of the total steps. Learning rate warmup is standard practice in LLM pre-training [32, 57, 42, 43, 58].

All models in the main experiments were trained on a Cerebras CS-3 system. 610M-parameter 20TPP models take roughly 6 hours each to train on a single CS-3.

For a given model configuration, we find results to be very stable across random seeds. To quantify the variance, we repeated 111M-parameter, 20 TPP training four additional times for six different hyperparameter settings, resulting in 5 total validation loss results for each of the six training runs. Standard deviation of the validation loss was below 0.003 in all cases.

**Proxy model hyperparameter tuning** To find the optimal $\mu$P hyperparameters (HPs), we trained a 39M proxy model using a width $d_{\text{model}}$ of 256, with 24 layers and head size of 64. We trained this model on 800M tokens with a batch size of 256 sequences and a context length 2048. We randomly sampled 350 configurations of base learning rates, base initialization standard deviation, and embedding and output logits scaling factors, and used the top-performing values as our tuned HPs (Table 4).

# D  Additional related work

## D.1  Optimizers for large-batch training

Prior work has explored optimizers designed specifically for large-batch training, including LARS [59] and LAMB [60]. It is instructive to consider these prior findings in light of the scaling laws from our paper. In particular, both original BERT [61] and the LAMB replication were trained on 85.2B tokens. Applying our fitted $B_{\text{crit}}$ power law over $D$=85.2B, we obtain an estimated $B_{\text{crit}}$ of about 12M tokens. Original BERT was trained for 90% of steps with a batch size of 65K tokens (512 sequences of length 128). LAMB increased the batch size to 4M tokens (32K sequences), justifying their claim, "BERT training can be reduced from 3 days to just 76 minutes" [60]. However, based on the predicted $B_{\text{crit}}$ of 12M, batch size 4M is still well within the expected range of efficient batch sizes. Moreover, the LAMB paper later notes, "we did not observe any speedup by increasing the batch size from 65536 to 131072 [sequences, or 16.8M tokens]." In other words, they reach the point of diminishing return *exactly where $B$ exceeds our predicted $B_{crit}$*.

It is likely that some optimization issues solved by LAMB (to enable stable large-batch training) are solved other ways in modern LLM training setups, via, e.g., gradient clipping, pre-LayerNorm placement, better initialization and stability control through $\mu$P, etc. Scaling $\lambda$ rather than $\eta$ with $B$, as we propose, further supports stable, efficient training. However, gradient redundancy imposes an inherent limit on useful batch sizes, ensuring critical batch size remains relevant.

## D.2 Critical batch size

Observations of critical batch size have previously been related to data complexity [62], loss curvature [39, 63], and model architecture [40].

Merrill et al. [64] define $B_{\text{crit}}$ as the largest $B$ such that loss does not degrade by more than a fixed fraction $\epsilon$ from the $B_{\text{opt}}$ setting. They measure this $B_{\text{crit}}$ instantaneously throughout training, by repeatedly branching from a checkpoint with different $B$ settings and assessing the impact on loss.

Recent work also defines $B_{\text{crit}}$ in terms of how $\eta$ scales with $B$ [31, 30]; unlike our work, these recent studies use a constant learning rate schedule and no weight decay.

We follow McCandlish et al. [11]'s definition of $B_{\text{crit}}$ (Definition 3.1). Given various theoretical assumptions, McCandlish et al. [11] derived a direct equivalence between $B_{\text{crit}}$ and what they call the *gradient noise scale* (GNS): the variation of the gradients between different training examples. However, they noted that the GNS "accurately predicts the largest usable batch size (at the order of magnitude level)," which is below the level of precision needed for large-scale training. Merrill et al. [64] recently found "the gradient noise scale underestimates the CBS [i.e., $B_{\text{crit}}$]." This lack of precision may be why, in Kaplan et al's original scaling laws paper [4], they note that, "although the critical batch size roughly matches the gradient noise scale, we are using a direct [empirical] measurement of $B_{\text{crit}}$." Our approach to measuring $B_{\text{crit}}$ (Sec. 3.2) similarly provides a direct empirical measurement, but one that can be efficiently computed with any learning rate schedule or optimizer.

## D.3 Detailed comparison with Zhang et al. [12]

Here we provide further comparison with the concurrent work by Zhang et al. [12]. The primary point of distinction of our paper is that we conducted a large-scale empirical study into the scaling of AdamW's weight decay hyperparameter (including its scaling with $B$), ultimately deriving a precise power law for the optimal AdamW timescale in tokens-per-parameter. Zhang et al. [12] did not use weight decay. Further, we also explored scaling of $B_{\text{opt}}$ in addition to $B_{\text{crit}}$. Beyond use of weight decay, further methodological differences in our main experiments include that we used a longer context length (2048 vs. 512), a cleaner dataset (SlimPajama vs. C4), the $\mu$P parameterization, a decaying LR schedule (more on this below), and that we tuned HPs at most $N$, $D$, $B$ (Table 3), while [12] performed a HP sweep for a 151M model, and re-used optimal values at other scales. We now focus on differences in estimating and measuring the scaling of $B_{\text{crit}}$.

**Estimating $B_{\text{crit}}$ for a specific target loss**   Both our work and Zhang et al. [12] require measuring, for different batch sizes, how many training steps it takes to reach a particular target loss. Since the number of steps to reach that loss is not known *a priori*, it is inherently difficult to study $B_{\text{crit}}$ when using a LR decay schedule, where you must specify the number of steps in advance. Using a constant LR (as was done in early work on $B_{\text{crit}}$ [11]) simply does not result in competitive models [29]. Unfortunately, it is not feasible to *search* for the precise step count needed, i.e., by conducting full training runs with different schedules/step budgets.

Zhang et al. [12] creatively solve this issue by conducting a single training run at a constant LR, while using weight averaging to generate higher-quality checkpoints for evaluation. With this approach, they still "need to frequently evaluate the model on a holdout evaluation set" [12].

Given LR decay, as opposed to weight averaging, remains the standard practice for current state-of-the-art LLMs, we independently developed a different approach. This led to the novel method described in our paper. In contrast with [12], we do not need to frequently evaluate the model, as we instead fit a $B$-specific loss power law through a few validation loss values (Sec. 3.2). Indeed, our approach may improve the efficiency of [12]'s method, as it would obviate the cost of continuous validation, which concerned them (see their section "Evaluation data size and frequency").

**Estimating the $B_{\text{crit}}$ power law**   Collecting $B_{\text{crit}}$ data across multiple model scales and loss targets is expensive. Zhang et al. [12] establish $B_{\text{crit}}$ scaling in $D$ through three targeted experiments:

- measuring $B_{\text{crit}}$ while scaling $N$ but leaving $D$ fixed to 3.07B
- measuring $B_{\text{crit}}$ while scaling $D$ but leaving $N$ fixed to 302M
- measuring $B_{\text{crit}}$ while scaling both $N$ and $D$ proportionally (at 20 TPP)

Interestingly, $B_{\text{crit}}$ was found to only scale weakly in $N$, but scale similarly whenever $D$ is scaled. They then fit a power law to their data points for the 302M-parameter model, obtaining the fit $B_{\text{crit}} = 22.91 D^{0.47}$ (in tokens).

In comparison, we took a more brute-force approach, computing $B_{\text{crit}}$ across many different $N$ and $D$ values, and ultimately fitting our $B_{\text{crit}}$ power law across multiple different model sizes and TPP settings (Fig. 1, *right*). Also, unlike [12], we assessed the quality of fit via computation of both $R^2$ values and parameter quantiles via bootstrapping (Sec. 3.3).

Recall also that [12] used a different definition of critical batch size. Let us denote their quantity $B_{\text{zhang}}$. They set $B_{\text{zhang}}$ to be the $B$ such that the data required to reach a loss target is $1.2 \times D_{\text{min}}$ (i.e., $1.2\times$ the data required with $B_{\text{opt}}$).

We can use Eq. (6) to align their fitted law with our own. By this equation, we have:

$$D = D_{\text{min}}(1 + \frac{B_{\text{zhang}}}{B_{\text{crit}}})$$
$$:= D_{\text{min}}(1.2)$$
$$\Rightarrow \frac{B_{\text{zhang}}}{B_{\text{crit}}} = 0.2$$
$$\Rightarrow B_{\text{crit}} = 5 B_{\text{zhang}}$$

Thus, to convert their coefficient to our scale, we multiply it by 5, and, dividing by the number of tokens in our sequences, obtain $B_{\text{zhang}} = 0.0559 D^{0.47}$. The $B_{\text{zhang}}$ coefficient (0.0559) is 19% larger than our own (0.0471), perhaps reflecting differences in training setup or data quality (and worth investigating further in future work). However, the exponents are quite similar (0.47 vs. 0.462), suggesting that both works are independently measuring the same fundamental scaling behavior.

We emphasize that $B_{\text{crit}}$ directly reflects the fundamental limit to data parallelism in training neural networks. Given the significant implications of $B_{\text{crit}}$ scaling in $D$ rather than $C$ or $L$ (including those discussed in Sec. 4), we note the scientific and practical value in having different approaches independently observe this same phenomenon.

### D.4 Hyperparameter scaling with $B$

It has long been recognized that the optimal learning rate, $\eta_{\text{opt}}$, scales with $B$, with reports of both linear [65–68] and square-root scaling [69, 60, 70]. Recent work has found $\eta_{\text{opt}}$ to *decrease* when $B > B_{\text{crit}}$ [30, 31], which resonates with our own findings (Fig. 2, *right*). Since it is difficult to predict exactly how $\eta$ will scale with $B$, studies of $B_{\text{crit}}$ have often done full HP sweeps at each $B$ [11, 40].

The only work we are aware of that specifically recommends scaling weight decay with $B$ is Loshchilov and Hutter [23], who suggest $\lambda \propto \sqrt{B}$, though this rule is not evaluated systematically. It is also important to note that Loshchilov and Hutter [23] use the *independent* form of weight decay, where decay is applied independently of the learning rate $\eta$, unlike common implementations such as AdamW in PyTorch [6]. In these more typical *dependent* implementations, weight decay is scaled by $\eta$, so any increase in $\eta$ with $B$ (e.g., $\eta \propto B$ or $\sqrt{B}$) already increases the effective weight decay strength accordingly.

### D.5 $\tau$ and effective learning rates

The concept of effective learning rates, influenced by weight decay, has been widely discussed [71–78]. In its simplest form, the effective or *intrinsic* LR is simply $\eta\lambda$, but in these prior works, effective LRs typically measure functional updates relative to weight magnitude, which is particularly relevant for normalization-based networks. Comparison of the effects of $\lambda$ vs. $\eta$ adjustments in the context of LR decay schedules was explored in [29].

The behavior of effective LRs (relative update sizes) over the course of training has been studied comprehensively by Kosson et al. [77], including comparing the effects of increasing $\eta$ vs. increasing $\lambda$. This work shows that higher $\eta$ values can cause large relative updates early in training, which can destabilize training or require longer warmups [58]. High $\eta$ and low $\lambda$ can also lead to larger weight norms [77, 78], which also has a destabilizing effect, particularly on low-precision training. These

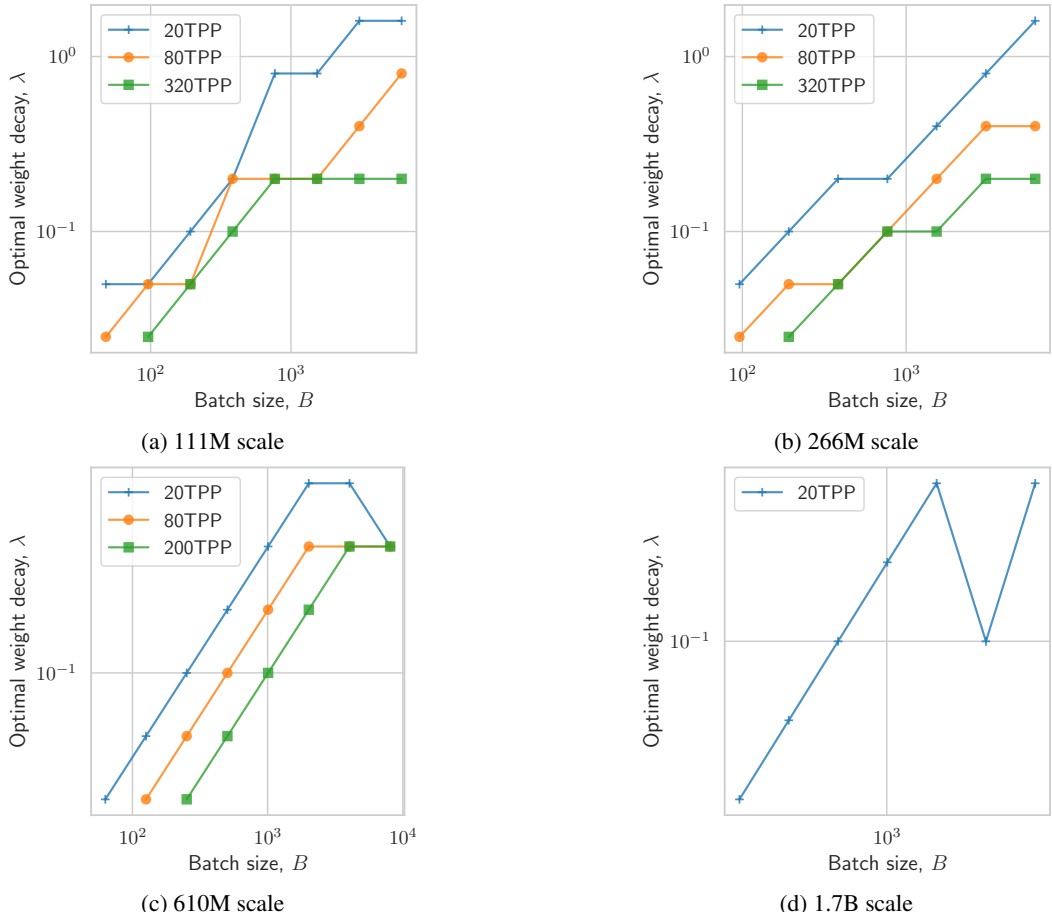

|             |             |
|:-----------:|:-----------:|
| (a) 111M scale | (b) 266M scale |
| (c) 610M scale | (d) 1.7B scale |

Figure 7: **Optimal weight decay scaling with** $B$: The optimal weight decay increases linearly over small batch sizes—until $B > B_{\text{crit}}$.

effects may explain why we were able to achieve higher effective LRs $\eta\lambda$ by tuning $\lambda$ rather than tuning $\eta$ with $B$ (Fig. 2, *middle*).

For a given dataset size $D$, the $\tau$ and the batch-normalized effective LR $\frac{\eta\lambda}{B}$ are equivalent, and thus effective LRs and the AdamW timescale can be viewed as different perspectives on the AdamW optimization process.

## E Scaling of $\tau$ and $\lambda$: additional details and results

### E.1 $\lambda$ scaling with $B$

Fig. 7 shows how optimal $\lambda$ changes across $B$, for all of the model scales and TPP levels where we did hyperparameter sweeps. There is a strong linear relationship between $\lambda$ and $B$ over the smaller batch sizes $B < B_{\text{crit}}$, with optimal $\lambda$ eventually plateauing (or decreasing). Note the standard use of $\lambda$=0.1 [3, 32–34] is only optimal at specific $B$, and this $B$ changes with TPP.

### E.2 Additional details on $\tau$ fitting

We now describe how we obtained the optimal $\tau$ values at specific model scales and TPP ratios. Rather than taking the empirical minimum loss, we fit a parabola to the $\langle L, \tau \rangle$ points in log space and took the analytic minimum of the parabola. If we have multiple loss values at the same $\tau$ (e.g., our data for a single scale and TPP comprises multiple different batch sizes), we only kept the lowest loss points at each $\tau$ prior to parabola fitting. We used validation loss on the held-out validation set. For

---

**Algorithm 1** Generating the optimal $\tau$ power law

---

**Input:** small batch size $B$, optimal per-$N$ learning rates $\eta$
Initialize $tau\_scaling\_law\_fitting\_points = [\,]$
**for** $N$ **in** $model\_scales$ **do**
   **for** $D$ **in** $10N, 20N, 80N, 320N, ...$ **do**
      Reset $loss\_points = [\,]$
      **for** $\lambda$ **in** $lambda\_range$ **do**
         Train $LLM(N, D, B, \lambda, \eta)$, get validation loss $L'$
         $\tau = {}^{B}/_{\eta\lambda D}$
         $loss\_points[\tau] = L'$
      **end for**
      $\tau_{\mathrm{opt}} = \arg\min_\tau (loss\_points)$
      $tau\_scaling\_law\_fitting\_points.add(\langle \mathrm{TPP}={}^{D}/_{N}, \tau_{\mathrm{opt}}\rangle)$
   **end for**
**end for**
Fit $c$, $m$ for $\tau_{\mathrm{opt}} = c\mathrm{TPP}^m$ on $tau\_scaling\_law\_fitting\_points$

---

our $\tau$ calculations, we input $B$ in units of *tokens* (in contrast to the *output* of our reported $B_{\mathrm{opt}}$ and $B_{\mathrm{crit}}$ scaling laws, which we report in units of sequences, of 2048 tokens). Algorithm 1 sketches the full procedure for generating the $\tau$ power law (Eq. (3)).

### E.3 Relationship to prior power laws

#### E.3.1 Relationship to $\eta_{\mathbf{opt}}$ scaling laws in dataset size, $D$

Both Shen et al. [16] and Bjorck et al. [22] propose scaling laws for the optimal learning rate, $\eta_{\mathrm{opt}}$, as a power law in the amount of data, $D$:

$$\eta_{\mathrm{opt}} = B \cdot c_{\eta_D} \cdot D^{m_{\eta_D}}$$

We now discuss how this power law also follows from the power law of $\tau_{\mathrm{opt}}$ in TPP. By Eq. (3), we have:

$$\tau_{\mathrm{opt}}(\mathrm{TPP}) = c_\tau \cdot \mathrm{TPP}^{m_\tau}$$
$$= c_\tau \cdot \left(\frac{D}{N}\right)^{m_\tau}$$

Substituting in the definition of $\tau$ (Eq. (2)), and assuming $\lambda$, $B$, and $N$ are fixed,[3] this implies $\eta_{\mathrm{opt}}$ will scale as:

$$\frac{B}{\eta_{\mathrm{opt}}\lambda D} = c_\tau \cdot \frac{D^{m_\tau}}{N^{m_\tau}}$$

$$\Rightarrow \eta_{\mathrm{opt}} = B\left(\frac{N^{m_\tau}}{\lambda \cdot c_\tau}\right) D^{-(m_\tau+1)}$$

$$= B \cdot c_{\eta_D} \cdot D^{m_{\eta_D}} \tag{11}$$

$$\text{where} \quad c_{\eta_D} = \frac{N^{m_\tau}}{\lambda \cdot c_\tau} \quad \text{and} \quad m_{\eta_D} = -(m_\tau + 1)$$

Eq. (11) is exactly the form of the power law used in Shen et al. [16], and explains the results across batch sizes seen in Bjorck et al. [22], as discussed in Sec. 2.4.

**Comparison to fit in Power Scheduler [16]** Given $c_{\eta_D} = \frac{N^{m_\tau}}{\lambda \cdot c_\tau}$ and $m_{\eta_D} = -(m_\tau + 1)$, we can use these formulas to compare our fit coefficients to those in Shen et al. [16].

In Shen et al. [16], they find $m_{\eta_D} = -0.51$. In our case, $m_\tau = -0.520$, and therefore $m_{\eta_D} = -0.48$, which is quite similar.

---

[3]Bjorck et al. [22] use a fixed $\lambda$=0.1, a fixed batch size of 0.5M tokens (for most of their experiments), and fit scaling laws separately at different model sizes (except in their Section 4). Shen et al. [16] use $\mu$P to adjust the LR for different model scales, so the derivation applies at any particular $N$; they do not report which optimizer is used nor any of its settings.

Comparing our $c_{\eta_D}$ to their $c_{\eta_D}$ ($= 4.6$) is a bit less straightforward. First of all, Shen et al. [16] inputs $B$ in sequences (of length 4096). We thus convert to the scale of our coefficient by dividing by their sequence length, obtaining $c_{\eta_D} = 0.0011$. Secondly, the power law of Shen et al. [16] is actually for the *base* $\mu$P learning rate $\tilde{\eta}$, while our derivation above assumes the adjusted (final) learning rate $\eta$ (Sec. 2.1).

Let us first compare $c_{\eta_D}$ coefficients at the proxy-model scale, i.e., where $\eta = \tilde{\eta}$. If we were to use a 28M-parameter proxy model, and a default $\lambda=0.1$ (and using our fit values of $c_\tau = 1.084$ and $m_\tau = -0.527$), then, by $c_{\eta_D} = \frac{N^{m_\tau}}{\lambda c_\tau}$, our $c_{\eta_D}$ would also equal 0.0011.

Now we consider how our coefficient varies when $N$ scales. To convert our $\eta$ scaling law into one for the base $\tilde{\eta}$, we can instead use $c_{\eta_D} = \frac{N^{m_\tau}}{\lambda \cdot \rho \cdot c_\tau}$, where $\rho = P/W$, $P$ is the width of the proxy model, and $W$ is the width of the target model. The width also affects the number of parameters, $N$, and hence the term $N^{m_\tau}$. In Transformers, $N$ scales roughly as $N \propto LW^2$, where $L$ is the model depth and $W$ is the model width. If we round the fitted exponent to $m_\tau \approx -0.5$, and substitute the value $\rho \propto 1/W$ into the denominator, we therefore have:

$$
\begin{aligned}
c_{\eta_D} &= \frac{N^{m_\tau}}{\lambda \rho c_\tau} \\
&\propto \frac{N^{-0.5}}{\frac{1}{W}} \\
&\propto (LW^2)^{-0.5} W \\
&\propto L^{-0.5}(W^2)^{-0.5} W \\
&\propto L^{-0.5}
\end{aligned}
$$

which is invariant to changes in $W$—i.e., the $\mu$P adjustment cancels out the model scaling in width. So, if we only scale $W$, $\tau$ scaling would stay in agreement with the Power Scheduler recipe, but if we increase depth, $\tau$ scaling would decrease $\eta$ proportional to $1/\sqrt{L}$ in a manner that is not accounted for in Shen et al. [16].

The key point is that the scaling law used by Shen et al. [16] is valid, indeed, has similar fitted exponents, to what would be predicted by the $\tau_{\text{opt}}$ scaling law—but in a specific context only (small models, or models only scaling in width). Moreover, we have shown it may be less effective to adjust $\eta$ in order to optimize $\tau$ (as these approaches implicitly do); we obtained superior results by instead adjusting $\lambda$. By considering $\eta$, $\lambda$, and $B$ holistically, our scaling laws are a superset of these laws for $\eta_{\text{opt}}$, as well as other laws that we discuss further presently.

### E.3.2 Relationship to $\eta_{\text{opt}}$ scaling laws in model size, $N$

Sec. 2.2 gave our recipe for tuning hyperparameters, for an arbitrary $N$, $D$, and $B$ setting. Here we advocated setting peak $\eta$ to the $\mu$P-adjusted learning rate (where the base learning rate comes from proxy-tuning). Rather than further adjusting this LR based on the dataset size or batch size, we argued for instead adjusting $\lambda$ so that $\tau$ is tuned to its optimal value. Based on both theory, and our empirical findings comparing tuning $\eta$ to tuning $\lambda$, we believe that using $\mu$P to scale $\eta_{\text{opt}}$ with model width is sufficient for well-tuned models. That is, the *theoretical* scaling law for $\eta_{\text{opt}}$ (in model width), given by $\mu$P, is sufficient for good performance. We discuss this perspective further in this section, specifically how the $\mu$P scaling law can explain recent work in *empirical* $\eta_{\text{opt}}$ power laws.

As noted in Sec. 2.1, when using $\mu$P, a base $\eta$ is tuned on a small proxy model, and then scaled depending on the width of the target model. Let $W$ be the width of the target model, and let $P$ be the width of the proxy model. $\mu$P prescribes scaling the optimal base learning rate, $\tilde{\eta}_{\text{opt}}$, down to $\eta_{\text{opt}} = \rho\tilde{\eta}_{\text{opt}}$, where $\rho = P/W$. That is, $\eta_{\text{opt}} = P\tilde{\eta}_{\text{opt}}/W$. As models grow in size, $P$ and $\tilde{\eta}_{\text{opt}}$ do not change, so $\eta_{\text{opt}}$ will scale $\propto 1/W$. Dey et al. [79, Figure 2] show that, indeed, a range of LLMs from the GPT, Llama, and DeepMind series are roughly following a scaling law where their chosen learning rate, $\eta$ is following $\eta \propto 1/W$. In other words, if one were to build a scaling law for $\eta_{\text{opt}}$ based on published LLM settings, it would roughly obey the $\mu$P theoretical scaling law.

Furthermore, we can develop a scaling law for $\eta_{\text{opt}}$ in model size, $N$, using $\mu$P, and show that it matches a recent empirical scaling law by Porian et al. [35]. The number of model parameters in any Transformer-based LLM scales roughly in the depth, $L$, and width, $W$, as $N \propto LW^2$. If we

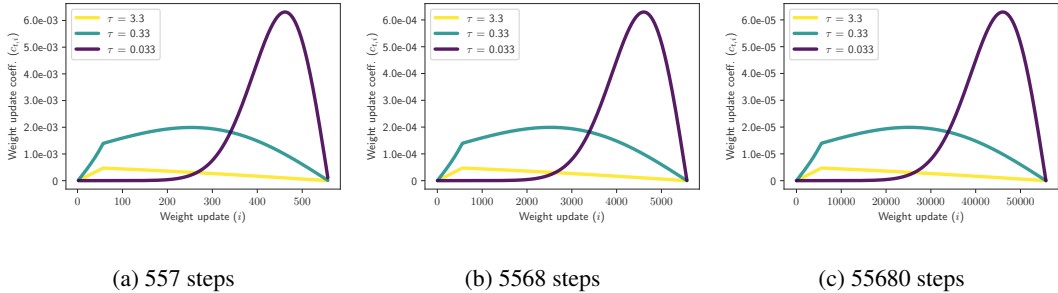

(a) 557 steps

(b) 5568 steps

(c) 55680 steps

Figure 8: $\tau$ **is invariant to steps, with a learning rate decay schedule** (111M scale, proxy-tuned peak $\eta$ with linear decay-to-zero): Here we adjust weight decay, $\lambda$, in order to maintain $\tau$ at a constant value, decreasing $\lambda$ proportional to the increase in $S$. Regardless of the total number of optimization steps, we see that the same $\tau$ corresponds to the same *shape* of the distribution of weight update coefficients (i.e., the same shape over the *data*, regardless of how the data is discretized in training). For batch sizing, this means that if we use a constant $D$ but increase $B$ by $K\times$ (decreasing $S$ by $K\times$), we will incorporate information across the data similarly—provided we use the same $\tau$.

assume that we maintain a fixed width-to-depth ratio, i.e., $R = W/L$, or $L = W/R$, then we have $N \propto W^3$, or $W \propto N^{1/3}$. Now, since $\mu$P prescribe scaling $\eta_{\text{opt}} \propto W^{-1}$, then for a fixed aspect ratio, $\eta_{\text{opt}} \propto N^{-1/3}$.

Taking a very different approach, Porian et al. [35] developed an empirical scaling law for $\eta_{\text{opt}}$ as a function of the number of model parameters. At each model scale, they trained with a variety of batch sizes and learning rates, and found the optimal settings of these hyperparameters. All models were trained to 20 TPP. They then fit a power law through the optimal LR settings, and found that $\eta_{\text{opt}} \propto N^{-1/3}$, exactly as would be expected if one simply followed $\mu$P.

As it provides a principled approach to scaling hyperparameters, $\mu$P can adapt to scaling when aspect ratio is not fixed. We therefore advocate using $\mu$P to set $\eta_{\text{opt}}$, rather than fitting special $\eta_{\text{opt}}$ power laws. However, with regards to our overall approach, it does not actually matter whether one uses the $\mu$P theoretical scaling law *or* an empirical one. The key point is that these laws can be used to set $\eta$ at a particular model scale, while the $\tau$ law should further be used to set $\lambda$ depending on the $B$ or $D$ values.

### E.4    The EMA perspective and learning rate schedules

To understand how the EMA view applies with a dynamic LR schedule, we follow the discussion of Bergsma et al. [29], who extended the formulation in Wang and Aitchison [1]. [29] consider EMAs with time-varying smoothing, $\alpha_t \in [0, 1]$. Letting $\alpha_1 = 1$ (i.e., $y_1 = x_1$), they express $y_t$ in terms of all inputs $x_t$:

$$
\begin{aligned}
y_1 &= \alpha_1 x_1, \\
y_2 &= (1 - \alpha_2)\alpha_1 x_1 + \alpha_2 x_2, \cdots \\
y_t &= \sum_{i=1}^{t} \left( \prod_{j=i+1}^{t} (1 - \alpha_j) \right) \alpha_i x_i
\end{aligned}
\tag{12}
$$

The EMA coefficient on each input is denoted $c_{t,i}$, where $c_{t,i} = \left( \prod_{j=i+1}^{t}(1 - \alpha_j) \right) \alpha_i$. In other words, $c_{t,i}$ reflects the contribution of input $x_i$ to output $y_t$ at time $t$, such that $y_t = \sum_{i=1}^{t} c_{t,i} x_i$. Unlike a standard EMA with a fixed smoothing parameter, in this extended EMA the coefficients need not decrease exponentially as $i$ decreases. Indeed, any set of coefficients can be generated by some particular smoothing schedule.

In terms of learning rate schedules for AdamW training, $\alpha_t = \eta_t \lambda$ becomes the smoothing parameter at step $t$ (cf. Sec. 2.1). The EMA itself, $y_t$, is the model parameters. The EMA is over weight updates: a large coefficient $c_{t,i}$ means the $i$th weight update contributes a lot to the EMA at step $t$. The EMA coefficients thus provide a more granular view of the contribution timescale than $\tau$ alone.

We now study the question, how do the EMA coefficients change as the step count changes? We generated the $c_{t,i}$ coefficients for the linear decay-to-zero LR schedule, and plot these coefficients at the final step (i.e., showing the contribution of weight updates to the final parameters). We use the $\mu$P-tuned and adjusted peak $\eta$, for 111M models. The learning rate increases linearly to the peak for the first 10% of steps, then decreases from the peak to 0 for the remainder of steps. We simulated three cases: where we take 557 steps, where we take 5568 steps, and where we take 55680 steps (5568 steps would be 20 TPP for a 111M model using $B$=192). From the perspective of batch sizing, these different steps could be achieved by decreasing the batch size twice by $10\times$.

We adjusted $\lambda$ for each step count in order to obtain the same three specific $\tau$ values. In Fig. 8, we see that the same $\tau$ implies the same *shape* of coefficients across the steps, and hence the same contribution over (normalized) time. That is, weight updates from the same portion of the training data contribute equally to the final model parameters.[4]

The key takeaway is that since $\tau$ is independent of the number of steps, it theoretically provides a $B$-independent measure of the AdamW timescale over weight updates, regardless of learning rate schedule. However, this equivalence for different $B$ breaks down when $B > B_{\mathrm{crit}}$ and weight updates themselves no longer contain linearly $B\times$ the information of a single sample.

# F    Scaling of $B_{\mathbf{opt}}$ and $B_{\mathbf{crit}}$: additional details and results

## F.1    Derivation of "extra data" Eq. (6)

Eq. (5) can be written as:

$$\frac{D - D_{\mathrm{min}}}{D_{\mathrm{min}}} = \frac{S_{\mathrm{min}}}{S - S_{\mathrm{min}}}$$
$$\Rightarrow (D - D_{\mathrm{min}})(S - S_{\mathrm{min}}) = D_{\mathrm{min}}S_{\mathrm{min}}$$
$$\Rightarrow DS - DS_{\mathrm{min}} - SD_{\mathrm{min}} = 0$$

Given $B = D/S$, we can substitute in $S = D/B$ to get an equation in a single variable, from which we can solve for $D$.

$$\frac{D^2}{B} - DS_{\mathrm{min}} - \frac{DD_{\mathrm{min}}}{B} = 0$$
$$\Rightarrow D^2 - DBS_{\mathrm{min}} - DD_{\mathrm{min}} = 0$$
$$\Rightarrow D(D - BS_{\mathrm{min}} - D_{\mathrm{min}}) = 0$$
$$\Rightarrow D = D_{\mathrm{min}} + BS_{\mathrm{min}}$$

Given $B_{\mathrm{crit}} = D_{\mathrm{min}}/S_{\mathrm{min}}$, we can substitute $S_{\mathrm{min}} = D_{\mathrm{min}}/B_{\mathrm{crit}}$ and obtain:

$$\Rightarrow D = D_{\mathrm{min}} + B\frac{D_{\mathrm{min}}}{B_{\mathrm{crit}}}$$
$$\Rightarrow D = D_{\mathrm{min}}\left(1 + \frac{B}{B_{\mathrm{crit}}}\right)$$

## F.2    Estimating $B_{\mathbf{crit}}$

We first provide some learnings from developing the $B_{\mathrm{crit}}$ estimation procedure.

First, regarding the functional form $L_B(D) = E_N + D_{\mathrm{const}}D^{-\beta}$, we found that including the irreducible loss term $E_N$ was important for obtaining good fits. $E_N$ conceptually represents the Bayes risk *plus* the minimum loss obtainable for a model of size $N$ (i.e., the first two terms of Eq. (8)). Second, only *interpolated* points were reliable; we only compute $B_{\mathrm{crit}}$ for loss values where all points are between, or very near to, curve fitting points. Third, each power law should have at least 3 points for fitting, in order to capture the concavity of the scaling in $D$. Finally, we sample our $B$ values logarithmically and, as in McCandlish et al. [11], perform our fits to Eq. (5) in log space.

---

[4]Note we do not plot the initial coefficients $c_{t,0}$ here, but they are equal across the scales for a given $\tau$, unlike the non-initial coefficients, which reduce by $10\times$ as the number of steps increases by $10\times$. So a constant $\tau$ means both the same contribution across the data, *and* the same *bias* (dependence on initial weights).

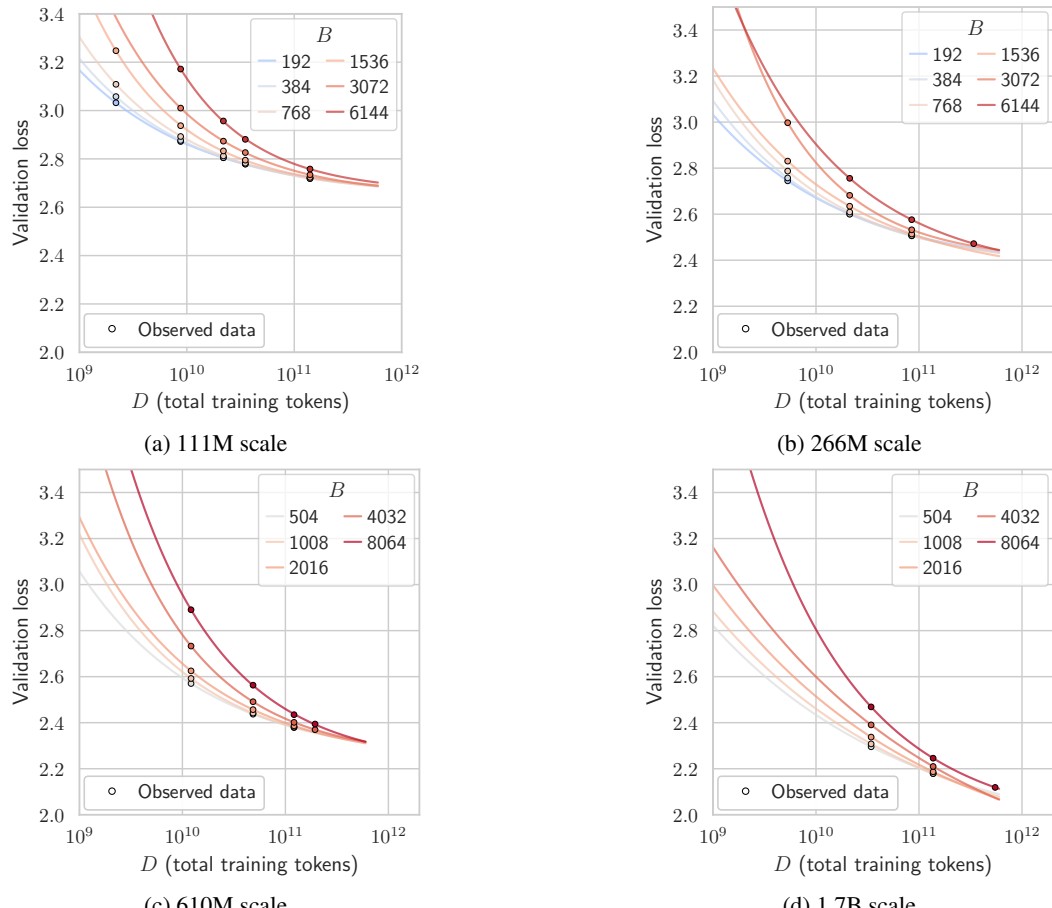

(a) 111M scale

(b) 266M scale

(c) 610M scale

(d) 1.7B scale

Figure 9: **Scaling laws in $D$ for computing $B_{\text{crit}}$**: Full set of $B$-specific power laws, for 111M to 1.7B scales, fitted after training models with different batch sizes, $B$, and dataset sizes, $D$, at each scale (empirical data from real training runs indicated by points). From these power laws, we can compute the amount of data needed to reach any target loss, as illustrated in main paper Fig. 3.

Fig. 9 illustrates all the fit scaling laws for the $B_{\text{crit}}$ experiments. Notice that beyond the fitting points, the curves may not predict behavior well. In particular, we would expect all curves to eventually converge as $D$ increases. Because loss targets beyond the fitting points are unreliable, we only compute $B_{\text{crit}}$ at loss targets where all the data sizes can be estimated through interpolation.

Fig. 10 shows the specific fits of Eq. (5) at particular loss targets. While Eq. (5) reflects the data trend well over the given $B$ values, we consistently find that points with very small $B$ do not approach $D_{\text{min}}$. We discussed this observation further in Appendix B.

Finally, for clarity, we provide Algorithm 2, which gives the detailed approach to generating the $B_{\text{crit}}$ power law in procedural form.

### F.3 Estimating $B_{\text{crit}}$ for the 3.3B model

To obtain an estimate of $B_{\text{crit}}$ for the 3.3B model (shown in Fig. 1, *right*), it was not feasible to apply our full $B_{\text{crit}}$ fitting procedure at this scale (i.e., fitting $B$-specific loss power-laws, etc.). Instead, we estimated $B_{\text{crit}}$ based on two $B$, $D_{\text{min}}$ pairs. That is, (based on an initial estimate of $B_{\text{crit}}$) we trained a 3.3B model to 23TPP with $B$=2016 and got a loss of 2.1688, and a separate model to 30TPP with $B$=4032, obtaining a loss of 2.1695. Given these losses are very close, these two models should have the same $B_{\text{crit}}$ and thus same $D_{\text{min}}$.

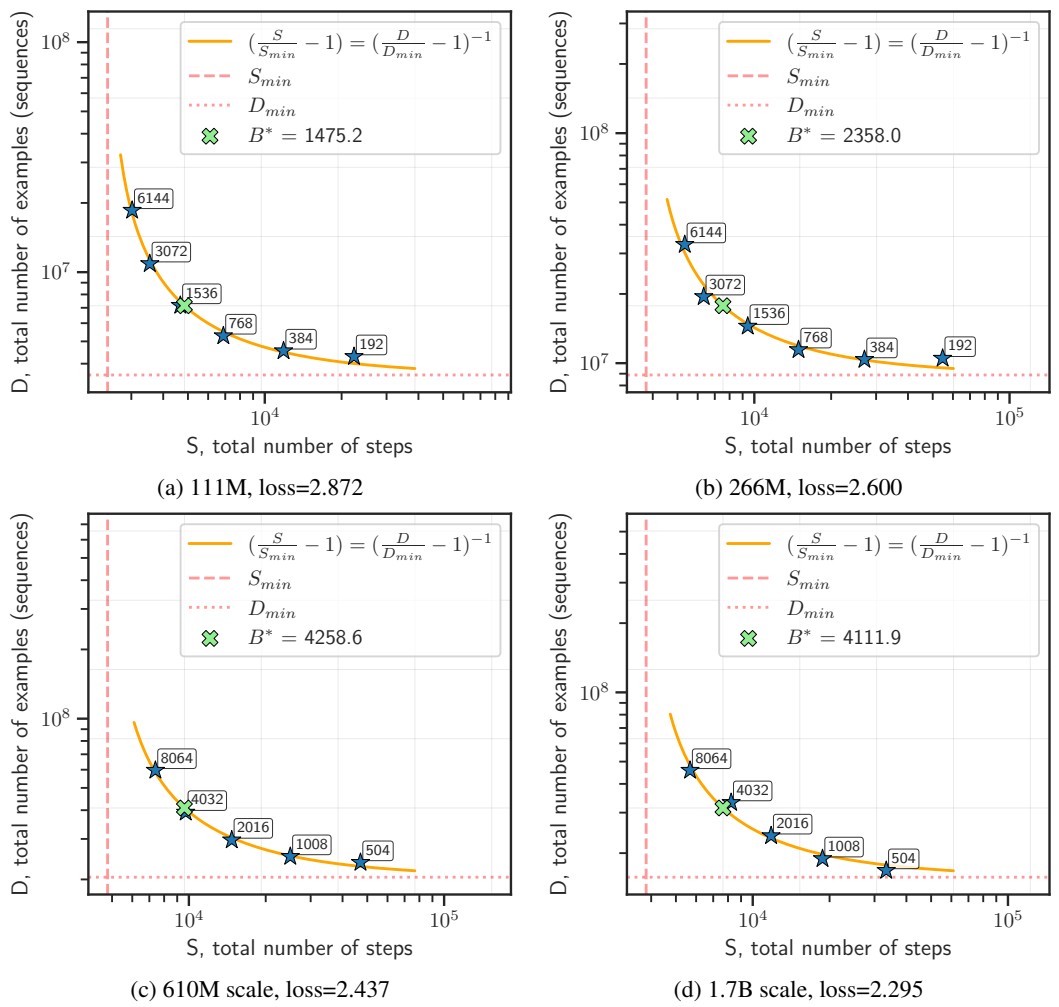

Figure 10: **Example fits of trade-off Eq. (5) plotting $S_{\mathbf{min}}$ and $D_{\mathbf{min}}$**: The empirical data has a good fit with Eq. (5) across model scales and loss targets. These estimates of $B_{\mathrm{crit}}$ are used in fitting the $B_{\mathrm{crit}}$-scaling power law (Fig. 1 *right*).

We solve for this $B_{\mathrm{crit}}$ as follows. Let $B_2 = 4032$ and $B_1 = 2016$. By Eq. (6):

$$D_2 = D_{\min}(1 + B_2/B_{\mathrm{crit}}), \text{ and}$$
$$D_1 = D_{\min}(1 + B_1/B_{\mathrm{crit}})$$

Let $r = D_2/D_1$ (i.e., 30/23 in this case). If we divide the above equations, and solve for $B_{\mathrm{crit}}$, we find:

$$B_{\mathrm{crit}} = \frac{B_2 - rB_1}{r - 1}$$

Plugging in our values of $r$, $B_1$, and $B_2$, we obtain an estimated $B_{\mathrm{crit}}$ of 4610, corresponding to a $D_{\min}$ of approximately 16 TPP.

### F.4 $B_{\mathbf{crit}}$ scaling in loss

Fig. 5 (*middle*) shows that $B_{\mathrm{crit}}$ is clearly not a power law in loss, as proposed in prior work [11, 4, 30]. However, if we only consider points with the same TPP, there does appear to be somewhat of a power-law relationship. In fact, this is implied by $B_{\mathrm{crit}}$ being a power law in $D$, along with the (standard) assumption that loss scales similarly in $N$ and $D$.

---

**Algorithm 2** Generating the $B_{\text{crit}}$ power law

---

Initialize $bcrit\_scaling\_law\_fitting\_points = [\,]$
**for** $N$ **in** $model\_scales$ **do**
   $\triangleright$ Fit $B$-specific (and $N$-specific) scaling laws, $L_B(D)$:
   **for** $B$ **in** $batch\_sizes[N]$ **do**
      Reset $scaling\_law\_fitting\_points = [\,]$
      **for** $D$ **in** $10N, 20N, 80N, 320N, ...$ **do**
         Train $LLM(N, D, B)$, get validation loss $L'$
         $scaling\_law\_fitting\_points.add(\langle D, L'\rangle)$
      **end for**
      Fit $E_N, D_{\text{const}}, \beta$ for $L_B(D) = E_N + D_{\text{const}}D^{-\beta}$ on $scaling\_law\_fitting\_points$
   **end for**
   $\triangleright$ Use $L_B(D)$ to estimate $B_{\text{crit}}$ at various loss values:
   **for** $\hat{L}$ **in** $lossTargets[N]$ **do**
      Reset $tradeoff\_fitting\_points = [\,]$
      **for** $B$ **in** $batch\_sizes[N]$ **do**
         Get $D_B = L_B^{-1}(\hat{L}) = \left(\frac{D_{\text{const}}}{\hat{L}-E_N}\right)^{\frac{1}{\beta}}$
         $tradeoff\_fitting\_points.add(\langle D_B, S={}^{D_B}/_B\rangle)$
      **end for**
      Fit $D_{\text{min}}, S_{\text{min}}$ for Eq. (5) on $tradeoff\_fitting\_points$
      $B_{\text{crit}} = {}^{D_{\text{min}}}/_{S_{\text{min}}}$
      $bcrit\_scaling\_law\_fitting\_points.add(\langle D_{\text{min}}, B_{\text{crit}}\rangle)$
   **end for**
**end for**
Fit $c, m$ for $B_{\text{crit}} = c(D_{\text{min}})^m$ on $bcrit\_scaling\_law\_fitting\_points$

---

Specifically, let $\hat{r} = D/N$ be the fixed TPP ratio. Therefore, $N = D/\hat{r}$. Assuming loss follows Eq. (8), we have:

$$L(N, D) = E + N_{\text{const}}N^{-\alpha} + D_{\text{const}}D^{-\beta}$$
$$= E + N_{\text{const}}\left(\frac{D}{\hat{r}}\right)^{-\alpha} + D_{\text{const}}D^{-\beta}$$
$$= E + N_{\text{const}}\hat{r}^{\alpha}D^{-\alpha} + D_{\text{const}}D^{-\beta}$$

Now, if $\alpha \approx \beta$, as is commonly accepted [3, 9, 80, 35, 81], we have:

$$L(D) = E + (N_{\text{const}}\hat{r}^{\alpha} + D_{\text{const}})D^{-\alpha}$$
$$= E + K_{\text{const}}D^{-\alpha}$$

where $K_{\text{const}} = N_{\text{const}}\hat{r}^{\alpha} + D_{\text{const}}$ is a constant. In other words, at a fixed TPP, loss is a power law in data. Given $B_{\text{crit}}$ is also fundamentally a power law in data, then by the transitivity of power law relationships, $B_{\text{crit}}$ is also a power law in loss in this context. This relationship can also be derived by expressing the $D$ in Eq. (7) as a power law in $B_{\text{crit}}$, and substituting into $E + KD^{-\alpha}$.

### F.5 Weight decay affects $B$ scaling laws

**Weight decay affects scaling of $B_{\text{opt}}$.** Table 5 illustrates how fixing $\lambda$ at different values alters the fitted $B_{\text{opt}}$ power law. Larger $\lambda$ systematically yields larger $B_{\text{opt}}$ and poorer fit quality (lower $R^2$). When $\lambda$ is fixed, the batch size that minimizes loss is only a *conditional optimum* (we denote it as $B_{\text{opt}|\lambda}$) because it compensates for suboptimal timescales $\tau$ rather than representing the globally tuned $B_{\text{opt}}$ obtained when $\lambda$ is optimized jointly.

The degraded $R^2$ values arise because a fixed $\lambda$ forces $B$ to balance two partially competing goals: (1) maintaining a good $\tau$ value, and (2) remaining below $B_{\text{crit}}$ to avoid gradient redundancy. A simple power law cannot capture this coupled behavior.

From the relation $\tau = B/(\eta\lambda D)$ and the empirical scaling $\tau_{\text{opt}} \propto ({}^{D}/_{N})^m$ (Eq. (3)), one can derive that, for constant $\eta$, $\lambda$, and $N$, the batch size preserving $\tau_{\text{opt}}$ should scale as $B \propto D^{m+1}$ (with $m + 1 \approx 0.47$ for our data). The exponents in Table 5, however, deviate from 0.47. For small $D$

Table 5: Different $\lambda$ settings systematically affect fitted power laws for $B_{\mathrm{opt}}$, and result in poorer fit quality (lower $R^2$). Fitted parameters change with $\lambda$ and consequently projected $B_{\mathrm{opt}}$ values (in sequences) for different token budgets $D$.

| Weight decay | Scaling law | $R^2$ | D=1e10 | D=1e11 | D=1e12 |
|---|---|---|---|---|---|
| 0.4 | $B_{\mathrm{opt}} = 0.0006D^{0.607}$ | 0.706 | 587 | 2377 | 9615 |
| 0.2 | $B_{\mathrm{opt}} = 0.0012D^{0.543}$ | 0.926 | 323 | 1128 | 3937 |
| 0.1 | $B_{\mathrm{opt}} = 0.0123D^{0.429}$ | 0.972 | 240 | 644 | 1729 |
| 0.05 | $B_{\mathrm{opt}} = 0.384D^{0.270}$ | 0.689 | 192 | 358 | 667 |
| 0.025 | $B_{\mathrm{opt}} = 10.3D^{0.120}$ | 0.161 | 163 | 215 | 284 |
| *Tuned* | $B_{\mathrm{opt}} = 0.0306D^{0.383}$ | *0.984* | *207* | *500* | *1207* |

and large $\lambda$, the $B$ that preserves $\tau_{\mathrm{opt}}$ lies near or above $B_{\mathrm{crit}}$, yielding worse loss due to gradient redundancy. This means $B_{\mathrm{opt}|\lambda}$ is artificially lower for small $D$ values. Since $B_{\mathrm{opt}|\lambda}$ is affected differently for different $D$, the scaling law slope is distorted (in this case, increased). Analogous issues disrupt $B_{\mathrm{opt}|\lambda}$ for small $\lambda$. These distortions reduce $R^2$ and impair generalization to large-scale training. When $\lambda$ is tuned, this confound is removed, and the resulting $B_{\mathrm{opt}}$ law aligns cleanly with the expected $D^{0.4}$ scaling.

**Weight decay affects scaling of $B_{\mathbf{crit}}$.** A similar confound arises for $B_{\mathrm{crit}}$. Fixing $\lambda$ causes deviations from $\tau_{\mathrm{opt}}$ to mingle with true gradient-redundancy effects. At 111M scale and a target loss of 3.03, e.g., larger batches perform worse solely because $\tau$ drifts from its optimal value, reducing the fitted $B_{\mathrm{crit}}$ from 867 (tuned $\lambda$) to 707 (fixed $\lambda=0.1$). At other loss targets, $B_{\mathrm{crit}}$ is less affected. Since the estimated $B_{\mathrm{crit}}$ is affected differently at different scales, the slope of the $B_{\mathrm{crit}}$ scaling law is again artificially distorted. $B_{\mathrm{crit}}$ will appear to increase faster than $D^{0.5}$, and projections to larger scales will be inaccurate. In contrast, tuning $\lambda$ isolates gradient-redundancy effects, yielding a stable $D^{0.5}$ relation that generalizes across scales.

