# OpenReview forum: "Power Lines: Scaling laws for weight decay and batch size in LLM pre-training"
_NeurIPS.cc/2025/Conference — NeurIPS 2025 poster_

### Official Review · Reviewer_f1ZL · 2025-07-02

**Clarity:** 3
**Significance:** 2
**Originality:** 3
**Rating:** 2
**Confidence:** 5

**Summary:**

This paper explores fundamental scaling laws for hyperparameters during large language model (LLM) pre-training, specifically focusing on two critical aspects: (1) optimal weight decay ($\lambda_{opt}$) and (2) optimal and critical batch sizes ($B_{opt}$, $B_{crit}$). The authors present extensive empirical studies conducted across multiple model sizes, dataset sizes, and batch sizes, leading to clear, well-supported scaling laws.

**Questions:**

1. Have you explored or considered how changes in numerical precision (e.g., training in mixed precision or using quantized formats) could affect your proposed scaling laws and recommendations? If so, could you briefly summarize your findings? If not, could you discuss how you expect reduced precision might influence batch size and weight decay scaling?

**Ethical Concerns:**

["NO or VERY MINOR ethics concerns only"]

**Final Justification:**

0. Sequence length is also quite important for LM pre-training, did you use a constant seq_len for all settings? how to ablate the impact of seq_len?

1. Does the proposed scaling law generalize to other optimizers like Muon?

2. What is the impact of different learning rate schedules beyond cosine decay with warmup? Have you ablated different warmup durations?

3. How does AdamW's epsilon parameter affect performance across different model sizes? This hyperparameter often requires adjustment as models scale.

I will be happy to increase my rate if the authors can address this.

**Limitations:**

1. **Generalizability of Method:**
    - *The proposed method for estimating $ B_{crit} $ is claimed to be suitable for arbitrary optimizers and learning rate schedules. Have the authors empirically validated or tested this methodology with optimizers other than AdamW (e.g., LAMB, Shampoo) or with alternative LR schedules (e.g., cosine decay, constant LR)? If not, what would be the expected challenges or limitations?*

2. **Practical Systems Implications:**
    - *Given that real-world model training is often heavily constrained by hardware limitations and communication overhead, how do the authors suggest practitioners incorporate these practical constraints when applying the proposed scaling laws and recommendations for optimal batch size?*

**Paper Formatting Concerns:**

Looks good to me.

**Quality:**

2

**Strengths And Weaknesses:**

### Strengths

1. **Strong Empirical Foundation:**
   The authors conducted extensive experimentation (around 400 models trained), covering a wide range of model sizes, batch sizes, and data scales. This experimental rigor strongly supports the reliability and generalizability of their findings.

2. **Novelty and Clarity of Contributions:**
   Introducing a clear, practical method for estimating $B_{crit}$—that is both efficient and generalizable—is a significant methodological contribution. The clear articulation of how to scale $\lambda_{opt}$ through the AdamW timescale $\tau_{EMA}$) is also valuable and original.


5. **Insightful Analysis and Theoretical Connections:**
   The authors provide intuitive explanations for observed phenomena, such as the EMA view of AdamW and its implications, and clearly demonstrate consistency with theoretical frameworks (e.g., maximal update parameterization (µP)).

---
### Weaknesses

1. **Limited Exploration of Optimizers and LR Schedules:**
   The main experiments focus exclusively on AdamW with a linear LR schedule. While the authors claim general applicability, additional validation with other optimizers (e.g., Shampoo, LAMB, Muon) and LR schedules (cosine decay, constant LR with averaging) would strengthen generalizability claims.

2. **Small Batch Behavior:**
   The observation that smaller batch sizes do not asymptotically approach $D_{min}$ as predicted theoretically is intriguing and warrants deeper exploration. Currently, explanations are only speculative.

3. **Practical Systems Considerations:**
   Real-world training is constrained by hardware limitations (e.g., bandwidth, memory, synchronization overheads). The paper does not significantly explore or discuss these practical system-level constraints and their impacts on batch size recommendations.

4. **Dataset and Architecture Limitations:**
   Experiments were conducted on just one dataset (SlimPajama) and one specific GPT-style architecture. While the authors provide a good range of sizes and scales, more diverse datasets, architectures, and domains might be needed to fully validate the universal applicability of these scaling laws.

---

> ### Author Rebuttal · Authors · 2025-07-29
>
> **Thank you for your helpful review.**  Your questions and comments will significantly improve our paper.  We particularly appreciate your recognition of the paper's strong empirical foundation and clarity of exposition, and your kind words about the significance, value, and originality of the methodological contributions.
>
> ### Regarding:  "Have you explored or considered how changes in numerical precision ... could affect your proposed scaling laws and recommendations?"
>
> This is a very interesting and timely question: Indeed, recent work (arxiv:2411.04330) showed that, in terms of scaling laws, lower precision reduces the model's "effective parameter count." Extending this idea, we hypothesize that:
> 1. There would be no impact on the scaling of Bopt or Bcrit, since these are independent of model size.
> 2. Lower precision would increase the "effective" tokens-per-parameter ratio (TPP) (via fewer effective parameters), thereby altering the optimal EMA timescale (Figure 1, left) and consequently the scaling of optimal weight decay.
> This merits explicit discussion in the paper and we will add it in the camera-ready version.
>
> ### Regarding: "The proposed method for estimating Bcrit is claimed to be suitable for arbitrary optimizers and learning rate schedules. Have the authors empirically validated this?"
>
> We agree that clarification is important here.  Our method in Section 3.2 is explicitly *designed* to be optimizer/schedule agnostic.
> - The B-specific power laws for *loss* (Figure 3) allow us to estimate Bcrit for a specific target loss without actually training to that loss.
> - The procedure applies directly to arbitrary schedules and optimizers as it fundamentally doesn't depend on multiple validation losses from a single training run (i.e., as when using a constant LR schedule).
>
> Thus, by design, the procedure itself does not strictly require empirical validation. However, you're correct that we have not yet applied the procedure beyond AdamW and linear decay. We acknowledge this limitation explicitly (Appendix B), and agree that exploring other optimizers is an important next step for our research.
>
> ### Regarding: "how do the authors suggest practitioners incorporate [hardware limitations and communication overhead] when applying the proposed scaling laws?"
>
> You raise a good point about practical system constraints.  While our paper emphasizes algorithmic scaling, we agree a more detailed discussion is beneficial.  In the camera-ready version, we'll clarify that our scaling laws explicitly define a practically relevant regime of training batch sizes (Bopt ≤ B ≤ Bcrit) (see our response to Reviewer SCxz for further details). Practitioners can leverage this identified regime alongside system-specific profiling (e.g., evaluating utilization at various batch sizes) to select optimal settings balancing algorithmic and hardware constraints.
>
> Moreover, we acknowledge that hardware limitations also impact the training time models in Section 4. Following Reviewer CcaT's suggestion, we are now exploring alternative, system-informed models of training time, and we plan to expand our discussion accordingly.
>
> ### Regarding the rating of the paper
>
> Given your many positive comments on the paper's empirical rigor, clarity, and methodological value, and the strong positive consensus among other reviewers, we respectfully ask if you'd reconsider your "Reject" rating. If our responses here clarify your concerns and our proposed improvements sufficiently address your highlighted limitations, revisiting your score could significantly impact both paper acceptance and potential spotlight considerations.
>
> We would be grateful for any further concerns or suggestions you might have. Thank you again for your thoughtful engagement and valuable insights.

---

### Official Review · Reviewer_CcAT · 2025-07-02

**Clarity:** 3
**Significance:** 4
**Originality:** 3
**Rating:** 5
**Confidence:** 4

**Summary:**

This paper conducts a large-scale empirical study on the interaction between weight decay $\lambda$, batch size $B$, token budget $D$, model size $N$, and learning rate $\eta$. At a high-level, the paper conducts hyperparameter scaling law analysis focused on the interaction of $\lambda$ with the other terms, encapsulated by the term $\tau_{EMA}$. On a more granular level, the paper's primary contributions are:

* The finding that the optimal value of $\tau_{EMA}$ obeys a power law decrease (as opposed to holding constant, as suggested by [Wang & Aitchison, 2024]).
* Methodology for estimating optimal and critical batch sizes, along with observed power law behavior of these parameters.
* Usage of the hyperparameter rules and scaling laws above to investigate pareto frontiers of FLOPs versus wallclock trade-offs. Notably, under a simple wall-clock model, the authors show that over-trained small models often dominate larger, Chinchilla-optimal models.

In short, the paper does as the authors claim in the abstract and introduction - empirical scaling law analyses of hyperparameters, especially batch size $B$ and weight decay $\lambda$.

**Questions:**

1. It is not immediately clear how the findings in Sections 3 & 4 are predicated on the weight decay analysis in Section 2. Do the authors have any sense (or data to back it up) that the findings on critical/optimal batch size and Pareto frontiers would change qualitatively had $\lambda$ been set via a simpler rule (such as the $\tau_{EMA} \approx c$ heuristic from [Wang & Aitchison, 2024])?

2. Would the Pareto frontiers change qualitatively when using a different mechanism to estimate critical batch size?

3. What's going on in the last few points of Figure 2, middle, where the optimal values of $\lambda$ and $\eta$ drop with the largest batch size?

**Ethical Concerns:**

["NO or VERY MINOR ethics concerns only"]

**Final Justification:**

I stand by every word in my review, and the other reviewer feedback seems to basically mirror my own. The author feedback was basically what I expected, so I have opted to keep my score as is.

**Limitations:**

Yes

**Quality:**

4

**Strengths And Weaknesses:**

## Strengths

I will be quite clear here - I think this is a good paper that significantly advances our understanding of hyperparameters in transformer pre-training. The paper is generally well-written, rigorous, explicit, and produces useful insights and analyses.

### Explicit methodology

Perhaps my favorite part of the paper is how explicit the methodology is. Consider Section 4.2. Rather than just giving the Pareto frontiers with some cursory discussion about their conclusions, the paper goes into detail about how it computed the exact Pareto frontier, (L251-256), with explicit references to what equations they used to fit what parameters. This makes the paper much easier to reproduce in other settings and on other models, and I think sets a great example to the scaling law community.

### Careful methodology

In addition to being explicit, the authors are clearly careful. This makes me trust their findings so much more. As a useful example (though this trend occurs throughout the paper) consider the estimation of critical batch size in Section 3.2. The authors identify a thorny issue (the fact that the number of steps, integral to the learning rate schedule, is unknown). They propose a specific method to remedy it, detail it explicitly, and discuss in even more depth in Appendix F.2. The authors have clearly thought about every aspect of their methodology, and I think the reader and community benefits from this kind of work being accepted.

### Breadth and utility of results

I think an undeniable strength of the paper is that it has a lot of useful results. Weight decay, batch size, and even what model size to train are all individually important topics. Including them all in the same paper is ambitious (though not without drawbacks, see below) but I think that the authors basically succeed on all fronts. It would be easy for practitioners to immediately incorporate this into their training regiments, and improve training on multiple fronts. The paper is also extremely useful in its breadth of connections to related work - someone relatively new to the area would get a pretty good understanding of the landscape of work just by reading this single paper.

## Weaknesses

I will go into a bit of detail here, because I want to emphasize ways the paper could potentially be improved. I want to be clear though, these are constructive suggestions, and I still wholeheartedly recommend acceptance.

### Breadth versus depth

My biggest critique of the paper is that the three main topics it covers (weight decay, batch size, pareto frontiers) are related, but not wholly so. This makes the paper a bit less cohesive. While I think the paper should be accepted as is, it makes me wonder if pulling the paper apart would have been at all beneficial.

To take this further, it means that each individual thrust has some empirical aspects and ablations that could have been expanded upon had it been the singular thrust of a paper. For example, one of the contributions in Section 3 is an estimate of the critical batch size. However, [McCandlish et al., 2018] provide a method for estimating this quantity as well. If Section 3 were the subject of a full paper, I would absolutely ask to see comparisons between the critical batch size estimates in this work, and the critical batch size estimates from [McCandlish et al., 2018]. In this paper, I don't think it's critical (given the breadth) but it's a comparison point that is missing.

Similarly, had section 4 been the primary focus, I would ask to see how the Pareto frontier changes as the training time model changes. This is done to some extent (Figure 6, left versus right) but there's a wide spectrum of possible models that would be fascinating to explore here. To some degree, this is a good criticism to have - it means that the paper is doing interesting things, and I look forward to follow-up work. But I do want to mention it as a drawback of fitting these three topics under a single paper's umbrella.

### Ablations and their role

Related to the above is the fact that by bundling these all together, it means that the paper is often combining multiple things into a single experiment, making it so that the findings are predicated upon one another.

As a concrete example, consider the critical batch size analysis in Section 3. It is unclear whether the optimal setting of $\lambda$ is actually required for any of the subsequent findings. Would the use of constant $\tau_{EMA}$ (as opposed to the power law decaying one in equation (4)) qualitatively change the findings in this section? It is essentially for the reader to know. This also rears its head in Section 4, which relies on the critical batch size estimates from this work. Do the pareto frontiers change significantly with different methodology for estimating the critical batch size?

I will note that these are not dealbreakers in any way shape or form. They can absolutely be investigated in follow-up work. But just as above, I want to note that there are trade-offs of the paper's design.

## Minor comments

* $\tau_{EMA}$ is undefined by the time Figure 1 shows up. More generally, I found Figure 1 to be much less useful as a synopsis of the work than the bullet points on L51-L55.
* In Figure 2, middle - you state that $\lambda$ scales linearly with $B$. While this is true for most $B$, this is clearly untrue for larger $B$. Should this be interpreted as $\lambda$ having a predictable relationship up to something like the critical batch size, or is something more complex going on?
* While I generally liked the colored text boxes with key takeaways, I found key takeaway 3 (L290) to be out of place and inappropriate for an academic setting. It's just a strange thing to advertise your methodology after you get done describing it. Moreover, it discusses the contributions in a pretty vague way that doesn't actually interact well with any of the specifics preceding it. I would strongly advocate removing it altogether.
* Figure 5, right, is difficult to match lines with the legend (e.g. I can barely tell which is 100M and which is 1B).
* Algorithm 1, in the appendix, mixes greek letters with pseudo-code names in a way that's very hard to read (e.g. $\tau_{EMA}$*_scaling_law_fitting_points.add*).
* The paper uses inconsistent \citet, \citep and the like. For example, L79 and L112 both use "Wang and Aitchison" in the sentence, but have different citation styles. This occurs elsewhere in the paper.

---

> ### Author Rebuttal · Authors · 2025-07-29
>
> **Thank you very much for your thoughtful and encouraging review.** We deeply appreciate your recognition of the paper's experimental rigor, writing, usefulness, and breadth of connections to related work, along with your constructive suggestions, which we address in detail below.
>
> ### Regarding "how the findings in Sections 3 & 4 are predicated on the weight decay analysis in Section 2"
>
> We did observe challenges in measuring Bopt and Bcrit prior to developing our weight decay scaling methodology, but we did not investigate this systematically until reading your points here.  Analyzing our data now, we find a very compelling story regarding the importance of tuning weight decay.
>
> Prior work uses a fixed weight decay when measuring batch size scaling laws (e.g., λ=0.1 in arxiv:2401.02954, λ=1e-4 in arxiv:2406.19146).  Using a fixed weight decay will not only result in worse loss (Section 2), but will affect the accuracy of the scaling laws.  We propose to add a new Section 3.4 to the paper titled, "Weight decay affects batch size scaling accuracy."  We will demonstrate this finding in two subsections: "Weight decay affects scaling of Bopt" and "Weight decay affects scaling of Bcrit".
>
> #### "Weight decay affects scaling of Bopt"
>
> In terms of fitting a Bopt power law, rather than tuning λ at each batch size, we tested setting λ to different fixed values.  We find that different weight decay settings *systematically affect the fitted power laws for optimal batch size*.  In the following table, we show the fitted parameters for different weight decay settings, as well as the projected Bopt (in sequences) for different token budgets:
>
> | Weight decay | Scaling law | D=1e10 | D=1e11 | D=1e12 |
> |:----:|:-----------:|:-----:|:-----:|:-----:|
> |  0.4  | Bopt = 0.0006 D^0.607 | 587 | 2377 | 9615 |
> |  0.2  | Bopt = 0.0012 D^0.543 | 323 | 1128 | 3937 |
> |  0.1  | Bopt = 0.0123 D^0.429 | 240 | 644 | 1729 |
> |  0.05 |  Bopt = 0.384 D^0.270 | 192 | 358 | 667 |
> | 0.025 |  Bopt = 10.3 D^0.120  | 163 | 215 | 284 |
> | **Tuned** | **Bopt = 0.0306 D^0.383** | **207** | **500** | **1207** |
>
> "Optimal" batch size increases with λ because the EMA timescale $\frac{B}{\eta\lambda D}$ is the fundamental scaling variable: if λ is larger, then B must increase to maintain the optimal ratio.  If the timescale is instead maintained across batch sizes via tuning λ (final row), the optimal batch size follows more accurate scaling.
> - Given the timescale $\frac{B}{\eta\lambda D}$ scales with $\mathrm{TPP}^m$ (Equations 2 and 3 in paper), it can be shown that when η, λ, and model size are fixed, "optimal" B scales as $D^{m+1}$ (where $m+1 \approx 0.473$ for our data).  I.e., this is how B will scale in order to maintain the timescale.
> - The exponent deviates from this value in the fits above because, for small D and high λ, the "optimal" B is near or above Bcrit, and therefore obtain worse loss, so "Bopt" is artificially lower for small D values.  Since "Bopt" is affected differently for different D, the scaling law slope is distorted (in this case, increased).  Analogous issues disrupt "Bopt" for small λ.
> - Measures of scaling law accuracy are consequently worse for fixed-λ fits, especially at larger scales.
>
> In summary, without tuning λ, the scaling law fit for Bopt is affected by small-scale effects, and will therefore not generalize to large-scale training, *even if the fixed weight decay is maintained at those larger scales.*
>
> #### "Weight decay affects scaling of Bcrit"
>
> Similarly, not tuning weight decay leads to inaccurate estimates of Bcrit.  For example, at 111M scale and a loss target of 3.03, larger batch sizes obtain worse loss purely because they cause τema to deviate from the optimal timescale.  This makes Bcrit appear lower than it actually is (i.e., from true gradient redundancy): from 867 when λ is tuned to 707 when we restrict to λ=0.1. At other loss targets, Bcrit is less affected.  Since the estimated Bcrit is affected differently at different scales, the slope of the Bcrit scaling law is again artificially distorted.  Bcrit will appear to increase faster than $D^{0.5}$, and projections to larger scales will be inaccurate.
>
> ### Regarding "Would the Pareto frontiers change qualitatively when using a different mechanism to estimate critical batch size?"
>
> As we discuss below, recent work has found other mechanisms (such as GNS) may underestimate Bcrit.  Similarly, as noted above, if λ is not tuned, the slope of the Bcrit power law will change.  In either case, such changes will affect the Pareto frontier in Section 4.  To illustrate this, we will add an experiment to the paper where we vary the exponent in the Bcrit power laws and plot the changes to the Pareto frontier.  We find that as the exponent increases, and Bcrit increases, models of higher TPP move to the frontier.  E.g., at very low Bcrit sizes, 20TPP models dominate the frontier, while at the highest Bcrit, 200TPP and 600TPP models are Pareto optimal.  In this way, inaccurate estimates of Bcrit lead to incorrect Pareto frontiers; large-scale training decisions based on these frontiers could lead to suboptimal training durations and compute costs.
>
> Together with the points above, we can now link suboptimal λ tuning (Section 2), to poor batch size scaling laws (Section 3), to incorrect conclusions regarding time/compute Pareto optimality (Section 4).  These additions thus help unify the paper's contributions across sections.  Thanks again for your very helpful contribution here!
>
> ### Regarding "comparisons between the critical batch size estimates in this work, and the critical batch size estimates from [McCandlish et al., 2018]"
>
> Thanks also for raising this; we definitely should have included a discussion of the gradient noise scale (GNS) from McCandlish et al., 2018 (arxiv:1812.06162), and will remedy this in the camera version of the paper.  A recent paper by Merrill et al. (arxiv:2505.23971) compared the GNS estimate of Bcrit to empirical measurements based on sweeping batch size and found "the gradient noise scale underestimates the CBS."  In fact, the original McCandlish et al paper only noted that the GNS "accurately predicts the largest usable batch size (at the order of magnitude level)," which is far below the level of precision needed for large-scale training.  This lack of precision may be why, in Kaplan et al's original scaling laws paper (arxiv:2001.08361), they note that, "although the critical batch size roughly matches the gradient noise scale, we are using a direct [empirical] measurement of Bcrit."  Our approach to measuring Bcrit provides an efficient empirical approach that can be used with any learning rate schedule or optimizer.
>
> We have also now developed instrumentation to measure GNS during training, so direct comparison between GNS estimates and empirical measurements in the context of modern LLMs is something we can also pursue in advance of the camera-ready deadline.
>
> ### Regarding Section 4 and other models of training time
>
> Exploring more realistic Transformer time models would indeed be valuable. Figure 6 (right) was interesting as the extreme version of model parallelism, but this is not realistic for the reasons mentioned on lines 285-286 (Transformer's inherent sequential nature).  A more realistic model of Transformer training could still assume perfect parallelism within a layer, but account for the "critical path" of sequential operations across the forward and backward passes.  Including Pareto frontiers for different depths under such a model would be a great addition to the paper - thanks a lot for the encouragement here!
>
> ### Regarding "what's going on ... where the optimal values of λ and η drop with the largest batch size?"
>
> Yes, we do interpret λopt as having a predictable relationship with B up until the critical batch size, at which point gradient information stops scaling linearly with B (due to redundancy), and the timescale over data breaks down.  Fortunately, the predictable scaling overlaps with the batch size "regime of interest" in practice: Bopt ≤ B ≤ Bcrit - please see our response to Reviewer S6CK for more details.
>
> Regarding why λopt and ηopt drop with the largest batch size: this is also a good point, and we should have discussed this more in this section.  As noted in our response to Reviewer 4mAg, prior work by Li et al (arxiv:2405.14578) (for vanilla Adam) showed that a kind of "surge phenomenon" happens with the optimal learning rate, where it first increase with batch size up to Bcrit and then decreases when B > Bcrit.  It's definitely interesting that we observed this with ηopt despite using AdamW rather than Adam.
>
> Yet our observations of this phenomenon for λopt are a bit mixed: while it definitely levels off around Bcrit, we don't observe consistent drops (see appendix Figure 7).  Note the prior work investigated extremely large batch sizes, even up to 10x or 100x Bcrit, which was not practical at the scales that we trained.  But this could be pursued further in follow-up work.
>
> ### Regarding the "Minor comments":
>
> Regarding your suggestions for Figure 1, Key takeaway 3, Figure 5, Algorithm 1, and use of `\citet` vs. `\citep`:
> - Thanks for these careful observations. We'll make all suggested corrections and clarify figures, text boxes, citations, and algorithms accordingly.

---

> > ### Comment · Reviewer_CcAT · 2025-07-31
> > **Thanks for the response!**
> >
> > Just want to note that I appreciate the detailed response. This all makes sense to me, and your various clarifications and new details basically line up with what I would expect. I believe the score on my review continues to be correct, and reiterate that I believe the work should be accepted.

---

### Official Review · Reviewer_S6CK · 2025-07-02

**Clarity:** 3
**Significance:** 3
**Originality:** 3
**Rating:** 5
**Confidence:** 4

**Summary:**

This work empirically studies hyper-parameters scaling of large langage models. The first part focuses on how to scale weight-decay and learning rates for AdamW when batch scales. They generalize a formula for a quantity $\tau = B/(\eta\lambda D)$ that was formulated in a multi-epoch setting and shown to stay invariant for optimal training in that case. They show that this quantity remains approximately constant in optimal training in the one epoch training. Additionally, they show that the weight decay $\lambda$ is "easier" to scale than the learning rate, in that tuning $\lambda$ instead of $\eta$ brings lower losses and that it scales better with batch. Finally, they show that when accounting for token-per-parameter (TPP) ratio, $\tau$ is not constant and behaves as a power-law in TPP. The second part focuses on how do the optimal and critical batches scale. They define experimental set-ups to measure $B_{crit}$ and $B_{opt}$ and show that both of these evolve as power-laws in compute. Additionally, different TPP yield different power-laws for $B_{crit}$ and $B_{opt}$. Finally, they discuss implications of their finding on the impact of batch size on the optimal trade-off between compute and training time. In particular, they show that the usual $TPP=20$ tokens per parameters requires not too large batch for efficient training.

**Questions:**

- figure 2, right : It seems to me that there does not seem to be any trend in $\tau$ unless I'm mistaken? Could you comment on that?
- figure 2, left : the trend in $\tau$ being constant mainly happens at small batch scale. But aren't we mainly interested in how this variable evolves for large batch scale? As soon as B=8064, there seem to be some large discrepancies (factor 10 variations). On the other hand, line 188, you underline that another measurement fails at small batch scale. Could you comment on which regime is of interest for batch when studying scaling laws and link it to the regimes you study?
- Did you consider plotting figures 3 and 4 in a log-log axis to better see the power-law scaling as linear curves?
- figure 5 right: I'm not sure to understant if the blue and green curves are fitted on the experiments and if yes, which data points were used and why did you not add the data points on the figures? Also I was wondering which TPP-ratio was used? It seems an important parameter from figure 5 left.

**Ethical Concerns:**

["NO or VERY MINOR ethics concerns only"]

**Final Justification:**

I believe this work should be accepted.

The autors present various interesting empirical scaling results that could be useful to practicionners and contribute to shed some light in the broad landscape of scaling models.

The main weakness in my opinion was the absence of links between the sections in the paper on batch scaling and weight decay scaling but the authors presented new results in the rebuttal on the links between both.

**Limitations:**

Yes, the authors discussed limitations in appendix B.

**Paper Formatting Concerns:**

I didn't notice any paper formatting issue.

**Quality:**

4

**Strengths And Weaknesses:**

**Strengths**
- This work presents a lot of practical results on scaling of hyper-parameters that could be very useful to practicioners.
- Experiments are performed at reasonably large scales (up to 3B parameters models).
- They generalize previous formulas such as from [1] to account for the TPP ratio and show that the optimal $\tau$ depends on TPP.
- The paper is well written with clear formulation of results in color boxes.

**Weaknesses**
- Different results in this work seem unrelated to each other, outside the fact that they all study hyper-parameters scaling. For example, the weight decay and learning rate scaling for AdamW in section 2 seems independent from the batch scaling results in sections 3, 4.
- The empirical results do not always support the claims: for example, I was not fully convinced by the claim "Sweeping $\eta$ shows a trend towards an optimal $\tau$" in figure 2 by looking at the right figure.
- I'm not sure why the authors rounded the exponent values in key takeways 1 and 2 (0.5 and 0.4). The results are slightly different? (line 110, 196) Especially line 196 the authors claim to be very close from a previous number by showing two digits precision but round it in the takeaway.

---

> ### Author Rebuttal · Authors · 2025-07-29
>
> **Thank you very much for the detailed review** and for highlighting the scale of the experiments, the quality of the writing, and the work being very useful for practitioners.
>
> ### Regarding: "the weight decay and learning rate scaling for AdamW in section 2 seems independent from the batch scaling results in sections 3, 4."
>
> You raise a fair point regarding the perceived disconnect between Section 2 and Sections 3-4.  Prompted by your comment (and Reviewer CcAT), we have run additional experiments clearly demonstrating that improper weight-decay tuning significantly distorts batch-size scaling results (see our response to Reviewer CcaT for details).  Specifically, when weight decay isn't properly scaled, it systematically distorts the optimal and critical batch-size power laws, leading to inaccurate Pareto frontiers.  We propose adding a new subsection (Sec. 3.4: "Weight decay affects batch size scaling accuracy") to explicitly unify these findings.  This addition clearly ties together the paper's results and highlights the importance of our weight-decay scaling methodology.
>
> ### Regarding: "why the authors rounded the exponent values in key takeways 1 and 2"
>
> Thank you for pointing this out.  Similar to prior work (abs/2406.19146), we rounded the exponent values (e.g., to 0.5 and 0.4) intentionally for simplicity and readability, but we see this introduced confusion given the precision emphasized elsewhere in the paper.  To resolve this clearly, we'll adjust the key takeaways to report precise exponent values consistently throughout, ensuring alignment with the main text.
>
> ### Regarding: "there does not seem to be any trend in η [in Figure 2, right]"
>
> You are correct, this wasn't clear.  The plot only shows η implicitly, as we sweep η while holding B fixed in order to get the curves for each batch size.  The actual trend we highlight is the lower boundary of the convex hull over all the points in the plot (over all the curves).  This is a bowl with its minimum around τema=0.21.  The 5 smallest batch sizes achieve their minimum within 2x of this τema.  However, as B increases, the minimum drifts higher.  We attribute this to the fact that the best *timescale* is always around τema=0.21, but to achieve this optimal timescale when using a high B, you would need to use a very high η.  In practice, we can’t push the η this high - we observe a maximum η, above which training becomes unstable; above this value, loss spikes occur from which training does not recover.  We will clarify this in the revised paper.
>
> ### Regarding: "Which regime is of interest for batch size when studying scaling laws?"
>
> This is an important question that deserves clearer discussion in the paper.  We propose to add the following discussion under Finding 1 to clarify which batch size regimes are of primary interest when studying scaling laws:
>
> > As batch size increases, we find that τema_opt remains roughly constant up to a certain point, after which it begins to drift. The drift point corresponds to the critical batch size Bcrit, above which gradient information no longer scales linearly with batch size and diminishing returns set in (Section 3).
>
> > In general, LLMs typically train faster and utilize hardware better with larger B, but only up to Bcrit: beyond this point, much more data (and compute) is needed to obtain the same loss, without meaningfully reducing the total number of sequential training steps (the poor tradeoff for B > Bcrit is depicted in Figure 4).  Furthermore, Section 3 will show that there is also an "optimal" batch size, Bopt, below which loss is worse *and* utilization/parallelism suffer (because batches are small).  In practice, LLMs should therefore be trained in the regime Bopt ≤ B ≤ Bcrit.  Notably, this is precisely the range where we have shown weight decay to scale predictably with batch size; our findings therefore support the direct optimization of weight decay in the most practically relevant training regimes.
>
> We also note that what counts as a "large" batch can vary by context. For example, optimizers designed for "large-batch" training, such as LAMB, are often evaluated at batch sizes still below Bcrit. See Section D.1 for further discussion; we found that analyzing LAMB's results through the through the lens of our scaling framework was very helpful for understanding the relevance of Bcrit in practice.
>
> ### Regarding: "plotting figures 3 and 4 in a log-log axis to better see the power-law scaling"
>
> Yes, thanks for this, but note that because the power laws used in Figures 3 and 4 have "irreducible" terms (e.g., the E term in $L(D) = E + D_c D^{-\beta}$), they actually do not reduce to straight lines on a log-log scale, rather they have asymptotes (at the minimum achievable loss in Figure 3, and the minimum number of steps and tokens in Figure 4).  We'll clarify this briefly in the text.
>
> ### Regarding clarity on Figure 5
>
> We will definitely revise to make Figure 5 more clear in the paper.
>
> - We will clarify that the blue and green curves in Figure 5 (right) are based on the power law fits from Figure 1 (middle and right).  We'll add the exact fitted equations to the plot to make this clear, and we will also mark Bdeepseek = 0.292(C)^0.3271 (the fitted law from [7]).  We did not repeat the fitting points here because this plot serves to compare our correct batch-size scaling (in dataset size D) with the misleading DeepSeek scaling (in compute C, which introduces a false dependence on model size).
>
> - We will also clarify that Figure 5 (left) is exactly the same data as in Figure 1 (middle), while Figure 5 (middle) is the same data as in Figure 1 (right).  Scaling in D (Figure 1) is the fundamental scaling relationship: the optimal and critical batch sizes scale in the dataset size D regardless of TPP, model size, or loss.  It is only when using another (misleading) scaling factor such as C that TPP or model size appears to be important, as in these plots.
>
> - Finally, we will emphasize explicitly that despite relying on a misleading scaling variable (C), DeepSeek’s corresponding choices for batch-size happen to align with our recommended regime (Bopt ≤ B ≤ Bcrit, as noted above).
>
> Your detailed feedback has significantly improved the clarity and coherence of our results. Thank you again for your valuable insights and thorough review!

---

> > ### Comment · Reviewer_S6CK · 2025-08-03
> >
> > I would like to thank the authors for their detailed answer. I am satisfied with their answer and will maintain my score.
> >
> > Just a note regarding your answer *"but note that because the power laws used in Figures 3 and 4 have "irreducible" terms they actually do not reduce to straight lines on a log-log scale"*, I think it will just reduce to a curve with two straight asymptotics lines on a log-log plot, which in my experience is still how papers present their scaling result, because I think it should still be more visible than on a non log-log plot.

---

### Official Review · Reviewer_4mAg · 2025-07-03

**Clarity:** 3
**Significance:** 3
**Originality:** 3
**Rating:** 4
**Confidence:** 5

**Summary:**

This paper study scaling laws for many hyperparameters: formulas for how to scale hyperparameters as we scale model size N, dataset size D, and batch size B.

**Questions:**

The paper cites in [54] that a "surge" phenomenon has been reported in the Adam optimizer. Could this issue potentially undermine the validity of the conclusions presented in this work?

**Ethical Concerns:**

["NO or VERY MINOR ethics concerns only"]

**Limitations:**

See Weakness 2 and 3.

**Quality:**

3

**Strengths And Weaknesses:**

**Strengths**

1. The author conducted numerous experiments and summarized the scaling laws of parameters.
2. AdamW and these hyperparameters are crucial for the pre-training of LLM.

**Weaknesses**

1. The authors only conducted experiments on dense LLMs. Nowadays, sparse activation LLMs are more popular. The authors can extend their experience to the MoE scenario.
2. The author's experience focuses solely on the AdamW optimizer.
3. The authors only conducted experiments on LLM pre-training tasks, which generally only require one epoch of training. It is currently unknown whether the authors' findings are applicable to other tasks that require multiple epochs of training.

---

> ### Author Rebuttal · Authors · 2025-07-30
>
> **Thank you very much for your helpful comments**, and for generously noting the numerous experiments, and the importance of our work in hyperparameter-tuning for pre-training LLMs.
>
> ### Regarding: the "surge" phenomenon with the Adam optimizer
>
> We believe the surge phenomenon observed in Li et al (arxiv:2405.14578) [54] *supports* some of the empirical findings presented in our work.  We definitely should have discussed the connection to [54] further in the main body of the paper; we will address this in the camera-ready version.
>
> We did note in the appendix: "Recent work has found ηopt to decrease when B > Bcrit [54, 53], which resonates with our own findings (Figure 2, right)."  However, it would be better to introduce this point when Figure 2 is first introduced, in the main body of the paper, and to discuss it in more detail.  The fact we observe: (1) a similar surge with optimal weight decay (as opposed to optimal η), and (2) a surge in the optimal η when using AdamW (rather than vanilla Adam as in [54]), suggests the surge phenomenon may be more general than initially presented by Li et al.  Further analysis is definitely warranted here.
>
> Thank you for raising this point!
>
> ### Regarding: "The authors only conducted experiments on dense LLMs. Nowadays, sparse activation LLMs are more popular."
>
> We agree MoEs are very important frontier-scale models. However, we intentionally chose dense LLMs for several reasons:
>
> - Dense models have fewer confounding factors (e.g., routing strategy, number of experts), making them simpler and clearer to study, and for other researchers to reproduce and build upon.
>
> - Dense models serve as a critical baseline for extending insights to MoEs in practice.  In our own studies, algorithmic innovations (including hyperparameter scaling strategies) are typically validated on dense models first.
>
> - Dense models remain widely deployed in both research and practical scenarios due to their ease of training, predictable hardware utilization, simpler inference pipelines, and suitability for fine-tuning.
>
> In fact, in our ongoing experiments with MoEs, we find similar fundamental weight decay and batch-size scaling relationships.  However, achieving these results required additional methodological innovations (e.g., in parameterizations, routing, etc.) that go beyond the scope of the current paper.  Thus, while MoEs represent a valuable direction for future work, studying dense models here was a justified, intentional scope choice rather than a weakness.
>
> ### Regarding: "It is currently unknown whether the authors' findings are applicable to other tasks that require multiple epochs of training."
>
> You're correct: our study specifically focuses on the practically important setting of single-epoch LLM pre-training. Prior work (Wang and Aitchison, ICML 2025) has indeed noted differences when using multi-epoch training, possibly due to data repetition. Reconciling these differences by isolating the effects of repetition versus scale is an interesting follow-up direction. We will explicitly clarify this scope limitation in the revised paper.
>
> Thanks again for your insightful feedback!

---

### Official Review · Reviewer_rwA5 · 2025-07-06

**Clarity:** 4
**Significance:** 4
**Originality:** 3
**Rating:** 5
**Confidence:** 3

**Summary:**

This paper does a comprehensive empirical study over different hyperparameters (HPs) in AdamW. The authors provide scaling laws for weight decay as a function of parameter count, data count and batch size, using models parameterized in muP. The authors also propose a new method for measuring the critical batch size and the optimal batch size and show the dependence of these quantities on the total data, showing that they do not scale as a power law of the target loss or FLOPs.

**Questions:**

Is there a similar EMA view that could be applied for general optimizers (i.e. Muon, Shampoo etc) - particularly matrix-based optimizers?

**Ethical Concerns:**

["NO or VERY MINOR ethics concerns only"]

**Quality:**

4

**Strengths And Weaknesses:**

The main strength of this paper is in the thorough empirical evaluation and the obtained scaling laws as a guide for how practitioners should change hyperparameters when scaling up. Parameterizing the models in muP removes the confounding effects on the learning rate when scaling up N and the finding that scaling up weight decay is better than learning rate is very valuable. Highlighting the key findings and takeaways in each section leads to a clear exposition of the ideas.

The methodology for fitting B_opt and B_crit is also interesting. I believe that the limitation explained by the authors in section 3.2 regarding the learning rate scheduler could be mitigated by incorporating weight averaging and using a constant learning rate during training, then decaying at various time steps - similar to [1]. Do you believe this will have a major impact on the results?

I do not find any major weaknesses in this paper.

[1] Wen, Kaiyue, et al. "Understanding warmup-stable-decay learning rates: A river valley loss landscape perspective." arXiv preprint arXiv:2410.05192 (2024).

---

> ### Author Rebuttal · Authors · 2025-07-29
>
> **Thank you so much for your thoughtful review**, and for the positive comments about the paper's comprehensive empirical evaluation, clarity of exposition, and the significant value of the obtained scaling laws for practitioners.
>
> ### Regarding: "incorporating weight averaging and using a constant learning rate during training, then decaying at various time steps"
>
> You are absolutely right: our approach applies to any LR schedule, but requires training over different dataset sizes.  If one uses a constant schedule plus weight averaging, one could instead identify the training steps required to reach a target loss by continuously evaluating the averaged model during training.  Alternatively, one could use a warmup-stable-decay (WSD) schedule, with different decay points (arXiv:2404.06395,arXiv:2405.18392).
>
> The weight averaging approach was previously used in Zhang et al. (arXiv:2410.21676).  As we note on lines 621-626 of the appendix, that approach still has the significant overhead of frequently evaluating the averaged models until one is found that obtains the target loss.  Our approach (fitting a scaling law through a selection of evaluated loss points) offers an orthogonal efficiency benefit, and could in fact be combined with weight-averaging (or WSD+decay) in order to reduce the number of evaluations.
>
> As for how a different schedule might impact results: this is indeed an empirical question, but we do hypothesize that as long as some effective variance-reduction approach is used in optimization (e.g., weight averaging or LR decay) then Bcrit will maintain its fundamental scaling with dataset size.
>
> We will add clarification about these points to Section 3.2 of the main paper, integrating and clarifying the points that are now in the appendix.
>
> ### Regarding: "Is there a similar EMA view that could be applied for general optimizers (i.e. Muon, Shampoo etc) - particularly matrix-based optimizers?"
>
> Yes, this is an active area of interest for us.  Many optimizers can indeed be interpreted in terms of an EMA timescale, although the exact derivation depends on the optimizer's structure:
> - optimizers such as Sophia (arxiv:2305.14342, Algorithm 3) use weight decay in a form that can be directly rewritten as an EMA, as was done with AdamW.
> - the standard way weight decay is used with Muon also admits an EMA interpretation.  E.g., in the recent MuonClip optimizer from Kimi (github.com: MoonshotAI/Kimi-K2/blob/main/tech_report.pdf), the update on Line 6 of Algorithm 1 can be directly revised into an EMA form.
> - the EMA perspective should also hold when EMA-compatible optimizers are applied in a different weight basis, e.g., as in SOAP (arxiv:2409.11321), where AdamW is applied in Shampoo’s eigenbasis (arxiv:1802.09568).
> - more generally, it is possible to analytically derive a composite timescale for optimizers that involve multiple momentum terms or weight averaging (as in AdEMAMix, arxiv:2409.03137), following the approach in our Appendix E.4.  The same principles apply but the derivation process necessarily depends on the specific optimizer.
>
> We fully agree with your suggestion and intend to explicitly expand our discussion of these connections in the final paper. Moreover, experimenting with additional optimizers is an important next step we are actively pursuing.
>
> Thank you again for your valuable feedback!

---

> > ### Comment · Reviewer_rwA5 · 2025-08-04
> > **Reply**
> >
> > Thank you very much for your reply and explanations! I will maintain my score and would be happy to see this work accepted.

---

### Note · Authors · 2025-08-13

We thank all reviewers for their thoughtful and detailed evaluations, which consistently recognized the breadth and rigor of our experiments (~400 models across N, D, B), the practical importance of the findings, and the clarity of exposition. In discussion, we addressed all substantive concerns, adding experiments that unify the paper’s contributions. In particular, we showed how proper weight-decay scaling is essential for accurate batch-size scaling laws and for correct Pareto-frontier analysis. We also expanded discussion of broader applicability (other optimizers/LR schedules), related phenomena (surge phenomenon, gradient noise scale), scope choices (dense LLMs, single-epoch pre-training), and practical guidance (Bopt ≤ B ≤ Bcrit). Reviewers indicated their questions were resolved and voiced support for acceptance. We believe these clarifications and additions make the paper an even stronger contribution, offering clear, reproducible scaling laws for weight decay and batch size that can directly guide compute- and time-efficient LLM pre-training.

---

### Decision · Program_Chairs · 2025-09-17

**Decision:**

Accept (poster)

**Comment:**

The paper significantly advances the understanding of hyperparameters in transformer pretraining. It has explicit and careful methodology, particularly in computing Pareto frontiers and estimating critical batch size, both ensuring reproducibility and making it a valuable resource. The results are broad and highly useful for practitioners (and the NeurIPS community more generally), offering practical insights on scaling weight decay, batch size, and model size to improve training. The work also provides a comprehensive overview of related research.

A key achievement is the generalization of existing formulas to incorporate the token-per-parameter (TPP) ratio, demonstrating that optimal values are TPP-dependent. The paper is well-written, with clear explanations and effective highlighting of key findings. Experiments were conducted at reasonably large scales, involving models up to 3 billion parameters, which enhances the credibility of the findings. A particularly valuable insight is the observation that scaling up weight decay is more beneficial than scaling the learning rate. The proposed method for measuring optimal and critical batch sizes is also noteworthy.

The reviewers have highlighted some important minor directions of improvement, and the authors have suggested many concrete changes, including multiple new appendices. I’m sure it goes without saying, but please implement these changes for the camera ready version.